# Changes in cerebrospinal fluid proteins across the spectrum of untreated and treated chronic HIV-1 infection

**Zicheng Hu**[1], **Paola Cinque**[2,3], **Ameet Dravid**[4,5,6], **Lars Hagberg**[7,8], **Aylin Yilmaz**[7,8], **Henrik Zetterberg**[9,10,11,12,13,14], **Dietmar Fuchs**[15], **Johanna Gostner**[15], **Kaj Blennow**[9,10], **Serena S. Spudich**[16], **Laura Kincer**[17,18], **Shuntai Zhou**[17,18], **Sarah Beth Joseph**[17,18,19], **Ronald Swanstrom**[18,20,21], **Richard W. Price**[22‡]*, **Magnus Gisslén**[7,8,23‡]

1 Bakar Computational Health Sciences Institute, University of California San Francisco, San Francisco, California, United States of America, 2 Unit of Neurovirology, San Raffaele Hospital, Milan, Italy, 3 Unit of Infectious Diseases, San Raffaele Hospital, Milan, Italy, 4 HIV Medicine and Infectious Diseases, Poona Hospital and Research Centre, Pune, India, 5 Noble Hospital and Research Centre, Pune, India, 6 Ruby Hall Clinic, Pune, India, 7 Department of Infectious Diseases, Institute of Biomedicine, Sahlgrenska Academy, University of Gothenburg, Gothenburg, Sweden, 8 Region Västra Götaland, Sahlgrenska University Hospital, Department of Infectious Diseases, Gothenburg, Sweden, 9 Institute of Neuroscience and Physiology, Department of Psychiatry and Neurochemistry, the Sahlgrenska Academy at the University of Gothenburg, Gothenburg, Sweden, 10 Clinical Neurochemistry Laboratory, Sahlgrenska University Hospital, Mölndal, Sweden, 11 Department of Neurodegenerative Disease, UCL Institute of Neurology, London, United Kingdom, 12 UK Dementia Research Institute at UCL, London, United Kingdom, 13 Hong Kong Center for Neurodegenerative Diseases, Clear Water Bay, Hong Kong, China, 14 Wisconsin Alzheimer's Disease Research Center, University of Wisconsin School of Medicine and Public Health, University of Wisconsin-Madison, Madison, Wisconsin, United States of America, 15 Institute of Medical Biological Chemistry, Innsbruck Medical University, Innsbruck, Austria, 16 Department of Neurology, Yale University School of Medicine, New Haven, Connecticut, United States of America, 17 Department of Microbiology and Immunology, University of North Carolina, Chapel Hill, North Carolina, United States of America, 18 Lineberger Comprehensive Cancer Center, University of North Carolina, Chapel Hill, North Carolina, United States of America, 19 UNC HIV Cure Center, University of North Carolina, Chapel Hill, North Carolina, United States of America, 20 UNC Center for AIDS Research, University of North Carolina, Chapel Hill, North Carolina, United States of America, 21 Department of Biochemistry and Biophysics, University of North Carolina, Chapel Hill, North Carolina, United States of America, 22 Department of Neurology, University of California San Francisco, San Francisco, California, United States of America, 23 Public Health Agency of Sweden, Solna, Sweden

‡ These authors are joint senior authors on this work.
* richard.price@ucsf.edu

**Data Availability Statement:** The full dataset for this study is posted online to Dryad, 10.5061/dryad.x3ffbg7tv.

## Abstract

Using the **Olink Explore 1536** platform, we measured 1,463 unique proteins in 303 cerebrospinal fluid (CSF) specimens from four clinical centers contributed by uninfected controls and 12 groups of people living with HIV-1 infection representing the spectrum of progressive untreated and treated chronic infection. We present three initial analyses of these measurements: an overview of the CSF protein features of the sample; correlations of the CSF proteins with CSF HIV-1 RNA and neurofilament light chain protein (NfL) concentrations; and comparison of CSF proteins in HIV-associated dementia (**HAD**) and neurosymptomatic CSF escape (**NSE**). These reveal a complex but coherent picture of CSF protein changes with highest concentrations of many proteins during CNS injury in the **HAD** and **NSE** groups and variable protein changes across the course of systemic HIV-1 progression that included two common patterns, designated as *lymphoid* and *myeloid* patterns, related to principal

**Funding:** This study was supported principally by NIH/NINDS research grants R01 NS094067 (RWP) and 3R01 NS094067-05S1 (RWP) and by the Swedish state under an agreement between the Swedish government and the county councils (ALF agreement ALFGBG-965885) (MG). Previous research grants supporting CSF specimen collections and background data assays included: P01 MH094177 (RS), P01 DA026134 (RWP Project PI), R01 MH081772 (SSS), R01 NS043103 (RWP), R01 NS37660 (RWP), R21 MH096619 (RWP); R01 MH062701 (RWP), R01 MH096619 (RWP), R21-MH083520 (RWP); Sahlgrenska Academy at University of Gothenburg (project ALFGBG-11067) (MG); and the Swedish Research Council (project 2007-7092.3) (MG). The funders had no role in the study design, data collection, decision to publish, or publication of the manuscript.

**Competing interests:** HZ has served at scientific advisory boards and/or as a consultant for Abbvie, Acumen, Alector, Alzinova, ALZPath, Amylyx, Annexon, Apellis, Artery Therapeutics, AZTherapies, Cognito Therapeutics, CogRx, Denali, Eisai, Merry Life, Nervgen, Novo Nordisk, Optoceutics, Passage Bio, Pinteon Therapeutics, Prothena, Red Abbey Labs, reMYND, Roche, Samumed, Siemens Healthineers, Triplet Therapeutics, and Wave, has given lectures in symposia sponsored by Alzecure, Biogen, Cellectricon, Fujirebio, Lilly, Novo Nordisk, and Roche, and is a co-founder of Brain Biomarker Solutions in Gothenburg AB (BBS), which is a part of the GU Ventures Incubator Program (outside submitted work). These organizations and companies had no role in the study design, data collection, decision to publish, or publication of the manuscript. MG has received research grants from Gilead Sciences and honoraria as speaker, DSMB committee member and/or scientific advisor from Amgen, AstraZeneca, Biogen, Bristol-Myers Squibb, Gilead Sciences, GlaxoSmithKline/ViiV, Janssen-Cilag, MSD, Novocure, Novo Nordic, Pfizer and Sanofi. These organizations and companies had no role in the study design, data collection, decision to publish, or publication of the manuscript.

involvement of their underlying inflammatory cell lineages. Antiretroviral therapy reduced CSF protein perturbations, though not always to control levels. The dataset of these CSF protein measurements, along with background clinical information, is posted online. Extended studies of this unique dataset will supplement this report to provide more detailed characterization of the dynamic impact of HIV-1 infection on the CSF proteome across the spectrum of HIV-1 infection, advancing the mechanistic understanding of HIV-1-related CNS pathobiology.

## Author summary

We measured more than 1,400 proteins in 303 cerebrospinal fluid (CSF) specimens from a representative broad range of untreated and treated people living with HIV-1 infection (PLWH) along with uninfected controls who volunteered for our studies. The results defined a complex, but generally coherent set of changes in many immune-inflammatory proteins and several central nervous system injury biomarkers as systemic HIV-1 infection progressed. There was a marked increase in many of these protein biomarkers in individuals with HIV-associated dementia and neurosymptomatic CSF viral escape, two conditions distinguished by overt, and characteristically severe, neurological impairment. The resultant data and initial analyses advance the characterization and understanding of the evolution of HIV-driven neuropathogenesis. The large dataset from this study is posted online and available to other investigators for further analyses, extending the utility of these data and potentially aiding in developing future prevention, diagnosis and mitigation of the deleterious impact of HIV-1 infection on the brain.

## Introduction

Exposure of the central nervous system (CNS) to HIV-1 is a nearly universal facet of untreated systemic infection that is readily documented by analysis of cerebrospinal fluid (CSF) obtained by lumbar puncture (LP) [1–11]. During the earlier phases of chronic infection, CSF viral isolates are genetically similar to plasma viruses and adapted to infect CD4+ T-lymphocytes (R5 T-cell-tropic viruses) [12]. As a consequence of the migration of infected CD4+ T-cells from blood into the CNS, where they release virus [12–15], CSF HIV-1 RNA levels are typically maintained at levels about one-tenth of those in blood [5,11,16–18]. Anatomically, HIV-1 RNA detected in CSF during this period predominantly represents a sample of virus from the leptomeninges, the site of a clinically silent aseptic meningitis with mild CSF pleocytosis [19–21]. An exception to this 1:10 ratio of CSF to plasma HIV-1 RNA concentrations develops in untreated individuals with very low CD4+ T-cell levels (< 50 cells per μL) in whom the ratio is closer to 1:100 when CSF white blood cell (WBC) counts are low, consistent with the concept that in these groups CSF virus is largely associated with CD4+ T-cell trafficking that includes infected cells and that when the number of these cells is diminished in late systemic infection, the CSF HIV-1 levels also fall [3,4,6,11,17,22–24].

The CSF:plasma HIV-1 ratio is altered in the opposite direction in patients presenting with subacute HIV-1-associated dementia (**HAD**) in whom high CSF HIV-1 RNA levels are nearly equal to those of plasma as a consequence of local viral replication within the CNS parenchyma in association with multinucleated-cell HIV-1 encephalitis (**HIVE**) [25–29]. This is generally

caused by HIV-1 variants capable of infecting macrophages and related myeloid cells that express low cell-surface densities of CD4 receptors (M-tropic viruses) [30–39].

In addition to the hallmark immunosuppression that, in its severe form, underlies susceptibility to **HAD** and the major opportunistic complications defining the acquired immunodeficiency syndrome (AIDS), chronic HIV-1 infection is also accompanied by systemic immune activation that predisposes to an additional set of chronic co-morbidities [40–44]. CSF sampling in cohort studies shows that intrathecal immune activation and inflammation are also constant features of chronic CNS HIV-1 infection [45]. Moreover, the character of this inflammation changes over the course of untreated infection as blood CD4+ T-cells decline [45,46]. In a recent study measuring a limited panel of inflammatory biomarkers, we identified two common patterns of CSF biomarker changes as blood CD4+ T-cell declined during untreated infection in the absence of **HAD**. The first pattern was characterized by a rise and then a fall in lymphocyte-related inflammation as blood CD4+ T-cells decreased from >500 to below 50 cells per μL, exemplified by CSF concentration changes in CXCL10, TNF-alpha and MMP-9 [45]. We have referred to this 'quadratic' pattern (based on trends modeling using a quadratic equation) of CSF viral load change with CD4+ T-cell loss as the *lymphoid* pattern of biomarker change. The second 'linear' pattern (modelled with a linear trends equation), that we refer to as the *myeloid* pattern of CSF inflammation, was exemplified by CSF concentration changes in the macrophage inflammatory markers, CCL2 and CD163, that rose more gradually and steadily through the course of blood CD4+ T-cell decline to reach their highest levels when blood CD4+ T lymphocytes fell to <50 cells per μL in those without **HAD** [45]. We introduce these two terms because they serve as prototypes for similar patterns of CSF protein biomarker change noted in this analysis, as discussed later. Importantly, as previously reported [45], CSF inflammatory protein reactions are generally *compartmentalized*, relate to local inflammatory processes within the CNS (including the leptomeninges, perivascular tissues and CNS parenchyma), and most often evolve independently from the concentrations of the same protein biomarkers in the blood, though this is a complex issue because the cellular sources and their functional context is strongly influenced (or determined) by systemic immune status. As a result, CNS inflammatory responses may change in parallel with or diverge from their systemic counterparts but are likely largely produced within the CNS in response to local conditions. CSF in **HAD** patients (with the underlying multinucleated-cell form of **HIVE**) characteristically exhibits a broad inflammatory profile with marked elevation of the CSF biomarkers that participate in both the *lymphoid* and *myeloid* CSF patterns, augmented by additional inflammatory biomarkers that may show only limited change in the non-HAD untreated individuals across the broad range of blood CD4+ decline [45].

While CNS immune and inflammatory reactions to local infection may both aid in local viral control and facilitate viral entry, they likely also contribute to neuronal dysfunction and injury, either directly by effects on neurons or through indirect effects on uninfected myeloid cells and astrocytes by largely uncertain immunopathological pathways [47–49]. Though a number of putative mechanisms of CNS injury have been identified, the exact contributions of individual virus-encoded and inflammatory molecules to *in vivo* injury remain largely inferential [50–62].

In addition to revealing inflammatory changes, CSF analysis provides evidence of neuronal injury during HIV-1 infection that can now be usefully monitored by measuring neurofilament light chain protein (NfL) concentrations in either CSF or blood [63–65]. The highest CSF and plasma NfL levels in untreated people living with HIV infection (PLWH) are detected in individuals with HAD. Elevated levels are also observed in a subset of neuroasymptomatic individuals, particularly some with low CD4[+] T-cell counts, indicating subclinical axonal injury. These elevated CSF and blood NfL levels can also predict the subsequent development

of HAD, with examples of increased CSF levels as long as one to two years prior to overt dementia symptoms in patients developing HAD in the pre-ART era [63]. While elevations of this axonal protein are not specific to **HIVE** [65], in the appropriate clinical context they usefully assess ongoing *disease activity*, and complement clinical examinations and neuropsychological test evaluations of cognitive-motor impairment that may be confounded by the cumulative effects of inactive (legacy) HIV-1-related and non-HIV-1-related CNS injury [66–68]. Even though HAD is uncommon in treated HIV-1 infection, mild neurocognitive impairment can sometimes be found in patients on ART. In these cases, blood NfL may serve as a useful marker to help discriminate between active ongoing neuronal injury and inactive, legacy CNS injury, whether due to HIV-1 infection or to other causes.

Combination antiretroviral therapy (ART) has a major impact on CNS infection, so that effective systemic viral suppression usually also induces similar CSF HIV-1 RNA reduction to below the clinical levels of detection [16,69–71]. Indeed, even when assessed by more sensitive 'single-copy' assays, ART that controls plasma viremia usually suppresses CSF HIV-1 RNA to very low or undetectable levels [71,72]. Interestingly, even in the presence of systemic treatment failure, CSF HIV-1 RNA levels usually are maintained at one-tenth or, more characteristically, even lower proportions in relation to those of plasma [17,70]. Moreover, systemically effective ART can prevent development of HAD and usually has a salutary effect on newly-presenting HAD, curtailing clinical progression and inducing variable clinical improvement, particularly when adjusted to deliver an effective combination of drugs to the CNS site of infection [73,74]. Monitoring reduction of CSF NfL can also be used to confirm the effectiveness of ART in preventing and mitigating CNS injury [68].

CSF/CNS inflammation accompanying local HIV-1 infection in the meninges and brain parenchyma is also reduced by effective ART. This includes resolution of CSF pleocytosis and reduction in the levels of various inflammatory biomarkers [17]. The dynamic effect of ART on inflammatory biomarkers has been nicely shown by rapid reduction in levels of CSF neopterin, an extensively studied inflammatory biomarker in HIV-1 infection, after treatment initiation [75,76]. Similar overall therapeutic effects have been documented with other inflammatory biomarkers [46,77–82].

However, CNS inflammation may not always fully subside to normal levels after suppressive treatment. For example, CSF neopterin, immunoglobulin production and T-cell activation may persist at levels above those of HIV-1-uninfected controls despite undetectable CSF HIV-1 RNA [83–86]. Even very low levels of CSF HIV-1 RNA, below the clinical cutoff of 'undetectable', may be associated with higher levels of local inflammatory activity than more complete CSF viral suppression [72,76,84,87,88]. Although the factors driving persistent inflammation are not firmly established, a CNS HIV-1 reservoir may be an underlying factor [15,89–93]. Whether this persistent inflammation is harmful or protective (or both) remains to be fully defined.

There are important, more substantial exceptions to the general CSF/CNS virological treatment efficacy in the setting of systemic viral suppression. These are referred to as *CSF HIV-1 escape* syndromes in which CSF HIV-1 RNA concentrations exceed those of blood, with the levels of virus in the blood generally suppressed or substantially reduced by the therapy [94]. They have been classified into three distinct types: *neurosymptomatic*, *asymptomatic*, and *secondary* CSF escape (abbreviated here as **NSE**, **AsE** and **2ryE**, respectively) [94–100]. **NSE** is the most important of these, since it is accompanied by neurological symptoms and signs, and in some patients by severe neurological injury [95,96,101–106]. Its neuropathology frequently exhibits a distinct variant of **HIVE** with prominent CD8+ T-lymphocytes [102]. Factors contributing to the development of **NSE** include reduced treatment adherence, drug resistance and component drugs in the ART regime with limited CNS penetration [103–105,107–110].

Fortunately, adjustments in ART, taking into account drug resistance, drug potency and CNS drug penetration, can usually halt disease progression and, to variable extent, reverse neurological deficits in many of these individuals [95,96], although for some **NSE** patients the course can be complicated by recrudescence [103,105,111]. In contrast, **AsE**, as the name implies, is not accompanied by neurological symptoms or signs, and likely is mostly transitory, perhaps akin to viral *blips* in plasma [112–114]. It may indeed be benign, though further studies of outcomes in these individuals are needed to more thoroughly characterize the natural history and clinical consequences of these episodes. **2ryE** is provoked by other (non-HIV-1) infections within the neuraxis in which the resultant inflammatory responses activate local HIV-1 production or replication. It characteristically resolves with effective treatment of the other infection [100].

Previous CSF studies of HIV-1 infection have focused on a number of the components of local CNS inflammation in diverse cohorts and case series, often emphasizing an individual stage of infection [17,54,75,115–122]. Most have also measured a relatively small number of inflammatory molecules [62]. For this reason, a more comprehensive picture of evolving CNS inflammation in relation to the stages of systemic infection, CNS injury, and treatment, has yet to be firmly defined [22,75,76,87,123].

The aim of this study, measuring a large number of CSF proteins in parallel, was to characterize the features of CNS inflammation and tissue responses more broadly as they change over the course of chronic HIV-1 infection in both untreated and treated persons. To this end we used the **Olink Explore 1536** platform to measure the concentrations of 1463 unique proteins in a total of 307 CSF samples, including 30 samples from HIV-seronegative controls and 277 samples from 272 PLWH (five individuals contributed a second CSF sample after their clinical status had changed during the cohort study in Gothenburg) representing a broad spectrum of clinical states defined by blood CD4+ T-cells, CSF HIV-1 RNA concentrations, treatment status, and clinically recognized neurological injury during chronic infection.

This report provides an introduction and initial overview of these identified CSF protein changes in relation to stages of systemic infection, local CNS infection, and CNS injury. Importantly, the online posting of the resultant dataset opens to other investigators the opportunity to extend the value of these data by exploring additional questions related to the CSF protein changes over the course of HIV-1 infection.

## Materials and methods

### Ethics statement

Ethical approvals were obtained from the institutional review boards of each center (Gothenburg Ethical Committee DNr 0588–01 and 060–18; UCSF IRB Protocol 10–0727; Milan, San Raffaele Scientific Institute IRB Protocol 235/2015; and Poona Hospital and Research Centre, Pune, India RECH/EC/2018/19/257). The samples from the cohort participants and the patients in the disease groups were obtained after written informed consent; if patients were unable to directly consent (some of those with **HAD** and **NSE**), this was obtained from a person with power of attorney or the equivalent.

### Study design and study subjects

This was a cross-sectional, exploratory study of proteins in a convenience sample of CSF specimens obtained in research studies at four clinical centers. These specimens were derived from existing study archives and represent 11 major clinical categories, along with a small number of additional samples from uncommon clinical settings that were not included in the main analyses in this report. The majority of CSF specimens were from longstanding cohort studies

at the University of Gothenburg in Gothenburg, Sweden [112] and the University of California San Francisco (UCSF) in San Francisco California, USA [17]. These cohort samples were supplemented by clinically-derived samples from ongoing HIV-1 studies in these two centers and selectively expanded by clinically-derived specimens from Milan, Italy [124] and Pune, India [125] representing the two neurological disease groups, *HAD* and *NSE*. Specimens were obtained between 1990 and 2020 within the context of research protocols. The sample groups included an uninfected control group and 12 chronically HIV-1-infected groups defined by blood CD4+ T-cell counts, neurological presentations, treatment status and viral suppression as previously described [123]. When available samples exceeded the planned number within each of these predefined groups, specimens were chosen at random, while in groups with sparse samples all available specimens were included. In brief, the following were the group criteria (the final specific individual group characteristics are described in the **Results and Discussion** section below).

## HIV-1 uninfected control (*HIV-*) group

These were drawn from the Gothenburg and San Francisco populations with similar demographic and background characteristics to the PLWH at those centers. Notably, the Gothenburg specimens were all selected from a group of individuals taking pre-exposure prophylaxis (PrEP) because of HIV-1 infection risk. As with the PLWH, all of these participants volunteered for the clinical examinations, background blood tests and the study lumbar punctures (LPs). All were screened with HIV-1 serology and blood HIV-1 RNA measurements.

**HIV-1-infected, untreated** included 7 groups of PLWH:

*Elite viral controllers* (*elites*) were defined by positive HIV-1 serology and undetectable plasma HIV-1 RNA concentrations on repeated occasions using a standard clinical cutoff (<20 HIV-1 RNA copies per mL) in the absence of ART [23,24,126].

*Five CD4-defined groups* were untreated (either naïve to treatment or off ART for at least 6 months) PLWH who did not have evidence (symptoms or signs) of active or confounding neurological disease on clinical examination. Following our experience in previous studies [45,127], CSF samples were segregated into five strata based on the accompanying blood CD4 + T-cell counts: >**500**, **350–499**, **200–349**, **50–199**, and <**50** blood CD4+ T-lymphocytes per μL, respectively. This division was based on previous experience in analyzing CSF biomarkers with emphasis on individuals with lower blood CD4+ T-cell counts since this is the range of vulnerability to *HAD*. None of the PLWH exhibited symptoms or signs of active or progressive neurological impairment based on bedside clinical history and physical examinations at the time of their visit. They were all characterized as *neuroasymptomatic* or exhibiting only chronic, stable neurological symptoms or signs. We did not attempt to segregate those with stable asymptomatic neurocognitive impairment (ANI) or minor neurocognitive disorder (MND) discerned by formal neuropsychological testing, but rather classified the subjects solely using the broad biological and diagnostic categories provided by the CD4+ group designations.

*HAD* patients were PLWH who presented clinically with new symptomatic, subacute or progressive neurological disease attributed to CNS HIV-1 infection by their caring physicians based on their clinical presentation and diagnostic evaluations. Some were studied before publication of the formal Frascati criteria for *HAD* [128] and diagnosed with AIDS dementia complex (ADC) stages 2–4 [129] while also meeting the American Academy of Neurology criteria for HIV-1-related dementia in place at the time [130]. Retrospectively, they also met the functional criteria for the Frascati diagnosis of *HAD* without the requisite formal neuropsychological assessment, and therefore this term was used to encompass the full group of these patients.

**HIV-1-infected, treated groups** included 5 main groups of PLWH taking ART.

Two groups of treated, *virally suppressed* individuals were selected from the Gothenburg and San Francisco cohorts: one with either normal age-corrected CSF NfL or previously unmeasured CSF NfL (*RxNFL-*), and a second unusual group of special interest selected on the basis of elevated age-adjusted CSF NfL (*RxNFL+*) levels when measured at the time of their cohort study evaluation.

*CSF escape* was diagnosed in individuals taking ART with suppression of plasma HIV-1 RNA below pretreatment levels but with higher concentrations of HIV-1 RNA in CSF than plasma [94,113,131]. Three types of CSF escape were distinguished: *NSE* in which individuals presented clinically with new moderate to severe neurological symptoms and signs that were attributed to CNS HIV-1 infection and without alternative cause after diagnostic studies; *AsE* were neurologically asymptomatic individuals identified by CSF and blood HIV-1 RNA findings during participation in cohort studies; and secondary CSF escape (*2ryE*) which was diagnosed on the basis of CSF and plasma measurements in the context of another, non-HIV-1 nervous system infection [100].

*Miscellaneous PLWH* included four anecdotally interesting individuals who did not fit the above categories: one who sustained HIV-1 cure (the 'Berlin patient') [132], two with very early initiation of ART [133], and one ambiguous escape patient with a history of progressive multifocal leukoencephalopathy (PML). These individuals were included in the full data table (**File 1 Dataset of CSF proteins in chronic HIV _infection_rev2024-09-10.xlsx** accessible on Dryad, DOI: 10.5061/dryad.x3ffbg7tv) [134] but not in the analyses described in this report which emphasizes group findings.

## CSF sampling

CSF from the cohort subjects was obtained according to previously described standard protocols [17,112,135,136]. Similar methods were used to process clinically-obtained samples from **HAD**, **NSE** and **2ryE** groups that were preserved for study purposes. In brief, CSF, obtained by LP, was placed immediately on wet ice and subjected to low-speed centrifugation to remove cells before cell-free fluid was either submitted directly to the Clinical Laboratory for HIV-1 RNA measurement along with plasma, or aliquoted and stored within 2 hours of collection for later HIV-1 RNA measurement along with stored plasma. Stored samples were maintained at ≤-70˚C until the time of biomarker assays or delayed HIV-1 RNA measurement.

## Clinical evaluations and background laboratory methods

Cohort subjects underwent standardized general medical and neurological assessments at study visits as previously described [137]. Symptomatic CNS disorders (**HAD**, **NSE**, and **2ryE**) were evaluated using standard clinical and neurological assessments, including neuroimaging along with CSF and blood measurements, to confirm their diagnoses and assess disease severity [137]. All cohort participants had routine clinical bedside screening for symptoms or signs of abnormalities in cognitive or motor function or evidence of CNS opportunistic infections or other conditions that might impact CSF biomarker concentrations. Individuals with CNS opportunistic infections or other conditions confounding these analyses were omitted with the exceptions of those in the *2ryE* group in which these infections were part of the clinical group definition. We initially set a general target of 25 separate samples for each major clinical group. Subsequently, some groups of particular interest were augmented with additional specimens, while for other categories there were shortfalls related to limited specimen availability.

HIV-1 RNA levels were measured in cell-free CSF and plasma at each site in local Clinical Research or Clinical Laboratories using the ultrasensitive Amplicor HIV-1 Monitor assay

(versions 1.0 and 1.5; Roche Molecular Diagnostic Systems, Branchburg, NJ), Cobas TaqMan RealTime HIV-1 (version 1 or 2; Hoffmann-La Roche, Basel, Switzerland) or the Abbott RealTime HIV-1 assay (Abbot Laboratories, Abbot Park, IL, USA), depending on the site and time period of collection. All recorded viral loads reported below 20 copies per mL were standardized to an assigned 'floor' value of 19 copies per mL for descriptive and computational purposes. Each cohort study visit included assessments by local Clinical Laboratories using routine methods to measure CSF white blood cell counts (WBCs), blood CD4+ and (in many) CD8+ T lymphocyte counts by flow cytometry, and CSF and blood albumin to assess blood-brain barrier integrity.

CSF NfL concentrations had been measured in multiple runs in a portion of the subjects prior to this study using a sensitive sandwich enzyme-linked immunosorbent assay (NF-light ELISA kit, UmanDiagnostics AB, Umeå, Sweden) [138] performed in the Clinical Neurochemistry Laboratory at the University of Gothenburg by board-certified laboratory technicians blind to clinical data; intra-assay coefficients of variation were below 10% for all analyses. To compare these NfL values across all individuals, we calculated age-adjusted NfL values for 50 years of age and compared them to the 50 year-old upper limit of normal of 991 ng/L, as outlined by Yilmaz and colleagues [68].

## Olink Explore methods

Samples from all sites were maintained at ≤-70˚C, aggregated in San Francisco, shipped together to Olink Laboratory in Watertown, MA, USA in previously stored, frozen aliquoted tubes, and analyzed in a single run in November 2020. Samples were transferred using local Olink procedures. The ***Explore 1536*** battery deployed at that time consisted of four plates: *Inflammation*, *Cardiometabolic*, *Oncology* and *Neurology* plates. The Olink immunoassays are based on the Proximity Extension Assay (PEA) technology [139], that uses a pair of oligonucleotide-labelled antibodies to bind to their respective target protein. When the two antibodies are in close proximity after binding, a new polymerase chain reaction target sequence is formed, which is then detected and quantified by Next Generations Sequencing (NGS). Output is reported in NPX units on a $\log_2$ scale of *relative concentrations* which are used for within-assay comparison of protein concentrations (https://olink.com/). Proteins measured using the **Olink** platform are identified in this study using the gene nomenclature and numbers as they appear in the UniProt database (https://www.uniprot.org/uniprotkb/) [140].

The level of detection (LOD) for each assay is listed in the data output and based on the background, estimated from the negative controls on every plate, plus three standard deviations. In the results presented below for this analysis, we include the returned values without adjustment or substitution when below the LOD. This is one of three strategies discussed by Olink and adopted for this study both for simplicity, and because it may increase the statistical power and provide a more normal distribution of the data without an increase in false positives (https://olink.com/faq/how-is-the-limit-of-detection-lod-estimated-and-handled/). Each protein measurement is comprised of four subgroups, each with a related LOD; these are listed in the full dataset now posted online (**File 1 Dataset of CSF proteins in chronic HIV _infection_rev2024-09-10.xlsx.** DOI: 10.5061/dryad.x3ffbg7tv) [134]. Our analysis also included measurements that were flagged in the data output file with quality control (QC) Warnings. These QC Warnings were generated if incubation Control 2 or Detection Control (corresponding to that specific sample) deviated by more than a pre-determined value (+/- 0.3) from the median value of all samples on the plate (https://olink.com/faq/what-does-it-mean-if-samples-are-flagged-in-the-qc/). Our own analysis of the effect of the QC warnings in the three proteins with quadruplicate repeats supports our decision to include these values as shown and is discussed in the context of **S1 Fig**.

## Statistics analysis

R version 4.1.1 was used for the main data analysis. The "hclust" function in R was used to perform hierarchical clustering to identify three clusters of the proteins. The "prcomp" function in R was used to perform the principal component analysis. One-way ANOVA was used to test the differential expression of proteins between the clinical groups. Biomarker associations were analyzed across the entire sample set using Spearman's rank correlation, graphically presented as a heat map for some. Comparison of biomarker concentrations between subject groups to address selected *a priori* questions used non-parametric methods including Mann-Whitney to compare two groups and Kruskal-Wallis with Dunn's *post hoc* test to compare three or more groups.

Two-sample t tests were used to compare the protein expression between the **HAD** and **NSE** groups. Partial correlation was used to assess the association between protein markers and the CSF HIV-1 level while controlling for the CSF NfL level (*NEFL* in Olink output). Similarly, partial correlation was used to assess the association between protein markers and the CSF NEFL level while controlling for the CSF HIV-1 level. Gene signature set enrichment analysis (GSEA) was used to calculate the enrichment score of the immune related pathways [141].

The immune related pathways were defined as the child terms of the term "immune system process GO:0002376" [142]. Because the **Olink Explore 1536** platform only measured a selected set of the human proteins, we only included an immune pathway if more than five of its proteins were measured by Olink assay. All statistical tests were adjusted for multiple testing using the FDR method.

Comparisons of the quadruplicate results of three proteins that were analyzed on each of the four plates and of earlier NfL measurements to the Olink NEFL results by linear regression used Prism 9.5.1 and 10.1, GraphPad Software, San Diego, California USA (www.graphpad.com). Selected comparisons of groups by ANOVA with correction for multiple comparisons and graphs of subject profiles were also produced using Prism 9.5.1 and 10.1 which was also used for redrawing some of the graphs produced by other methods.

## Dryad DOI

10.5061/dryad.x3ffbg7tv.

## Results and discussion

This is the initial report of a study using the **Olink Explore 1536** platform to measure CSF proteins in large sample of grouped specimens representing the spectrum of untreated and treated chronic HIV-1 infection. The resultant full study dataset (**File 1 Dataset of CSF proteins in chronic HIV _infection_rev2024-09-10.xlsx**) is posted online at *Dryad*: DOI: 10.5061/dryad.x3ffbg7tv (https://datadryad.org/stash) [134].

We have divided the presentation of the study results and discussion into two principal sections. The first, *Background* results section, describes the clinically-defined groups in the study cohort along with selected characteristics of the **Olink Explore 1536** data output that are illustrated in **S1** and **S2** Figs by analysis of three sets of quadruplicate-repeat measurements embedded in the Olink panel and measurements of seven CSF proteins that have served as biomarkers of different cellular components of CNS injury in previous studies. The second, *Main* results section, examines salient features of the full set of measured CSF proteins in three sets of analyses that focus on: **1**. The overall changes in proteins across the study groups; **2**. correlations of the measured CSF proteins with biomarkers of CNS infection (CSF HIV-1 RNA) and

CNS injury (CSF NfL); and **3**. comparison of CSF proteins in two clinical forms of **HIVE**: **HAD** and **NSE**.

## Background analysis of the study groups and selected features of the Olink Explore platform applied to the CSF samples

**Study group characteristics.** This cross-sectional analysis of a convenience sample of 303 CSF samples follows a strategy used in several of our previous studies of CSF biomarkers [45,75]. It draws on the experience and archived CSF specimens of four research centers with long-term interests in HIV-1-related CNS disease. The complete set of 307 analyzed CSF samples included 30 specimens from HIV-1-seronegative controls and 277 from 272 PWH (five PLWH contributed a second CSF sample after their clinical status had changed during the cohort study in Gothenburg) representing 7 untreated and 5 ART-treated groups that are the focus of this report, along with a miscellaneous group of four samples from individuals who do not clearly fit into any of the main categories and therefore are not included in this analysis which centers on group identities and differences. The background demographic and HIV-1-related clinical laboratory characteristics of the individual study participants, along with the **Olink** protein measurements of the entire set of 307 CSF samples, are all included in Dryad-posted dataset (DOI: 10.5061/dryad.x3ffbg7tv) [134].

The background features of the subject groups are summarized in **S1 Table** and graphically detailed in **Fig 1.** The latter shows the individual values for each of these background variables within the group designations. The main features of these subject groups are discussed in more detail in the legend to this figure.

Despite demographic imbalances in this convenience sample, as discussed in the legend to **Fig 1**, the group of CSF specimens provides a robust aggregate that encompasses the main facets of chronic, evolving HIV-1 infection in the absence and presence of treatment, and importantly includes two groups with HIV-1 encephalitis (**HIVE**): the untreated **HAD** and treated **NSE** groups. Here we briefly focus on selected background features that particularly bear on the main CSF protein findings and on some of the interpretive themes that follow. This includes introduction to the prototypes of the *lymphoid* (CSF HIV-1 RNA and CSF WBC counts) and *myeloid* (CSF neopterin) patterns of change across the five untreated, neurologically asymptomatic **CD4-defined** groups introduced earlier and defined in the **MATERIALS AND METHODS** section, along with the CSF NfL concentrations in a subset of the specimens that had been previously measured.

## CSF HIV-1 RNA concentrations and CSF WBC counts: <u>Lymphoid</u> patterns of change in the CD4-defined groups

As discussed in the legend to **Fig 1**, The concentrations of HIV-1 RNA in the blood and CSF, and hence their interrelationships, changed with progression of untreated infection in the **CD4-defined** groups (**Fig 1H–1J**). While in the blood there was a nearly steady, stepwise increase in HIV-1 RNA concentrations as the blood CD4+ T-cell count decreased from the highest (**CD4 >500**) to lowest (**CD4 <50**) group levels, in the CSF progressive viral load increase was interrupted in the **CD4 <50** group by a drop in the median HIV-1 RNA to 3.09 $\log_{10}$ copies per mL from a peak median of 4.14 $\log_{10}$ copies per mL in the **CD4 50–199** group, resulting in a high CSF-blood difference in this group (2.75 $\log_{10}$ copies RNA per mL shown in **Fig 1J**). This contrasted with the more constant differences between CSF and blood HIV-1 RNA of about 1.0 $\log_{10}$ copies per mL in the other four CD4-defined groups. These changes in CSF HIV-1 RNA were parallel to, and likely related to, the CSF WBC counts (**Fig 1K**) in which there was also a decrease in the **CD4 <50** group to negligible levels in comparison to the other

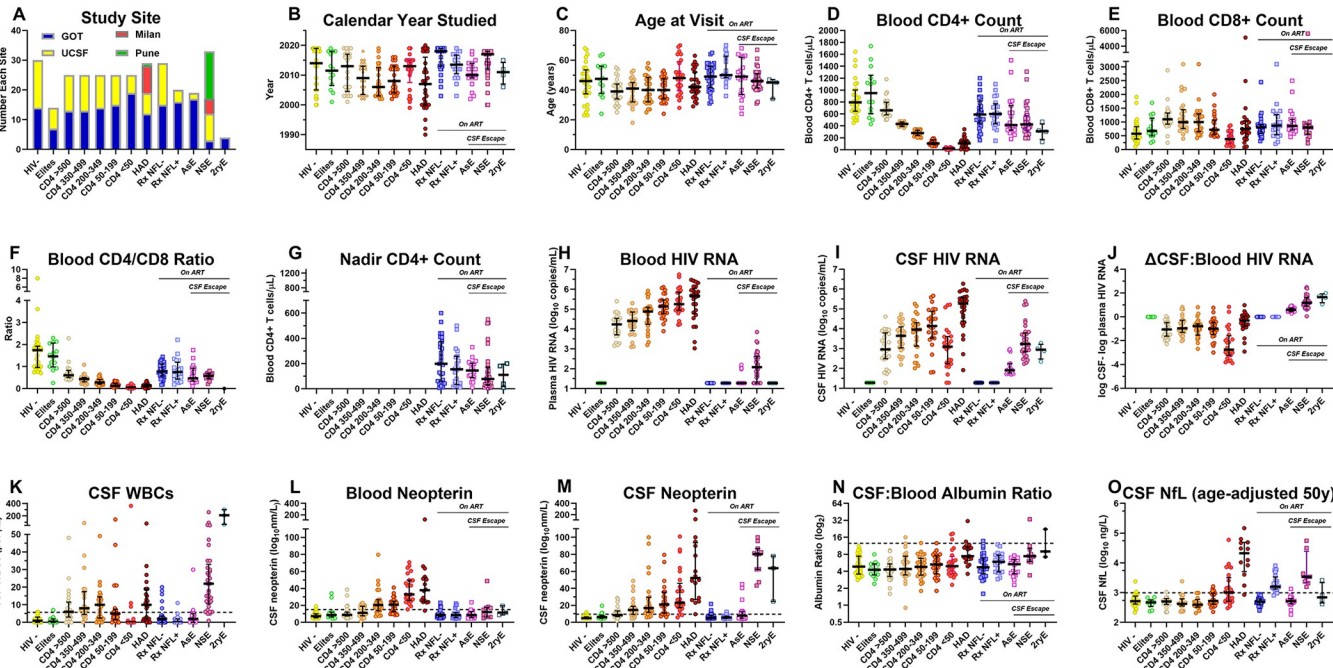

**Fig 1. Study group background characteristics.** Except for the first panel (**Fig 1A**), the graphs (**Fig 1B–1O**) use the same format that is applied to subsequent figures showing the Olink-generated protein measurement profiles across the study groups. This includes the symbols and the median and IQR bars. The dashed horizontal lines in **Fig 1K–1O** indicate the mean +2 SD of the uninfected, seronegative (**HIV-**) control group in the measured variable as a rough estimate of their upper normal limit and as a visual guide to compare groups; this convention is also used in the Olink CSF protein measurement figures showing the group profiles in subsequent figures. **A. Study sites.** Most of the samples were obtained in the context of scheduled outpatient visits during long-term cohort studies in Gothenburg, Sweden (GOT) and San Francisco, California, USA (SF) that included lumbar punctures (LPs) as part of the study protocols (**Fig 1A**). This included CSF specimens from the **HIV-**, elite controllers (**elites**), five neuroasymptomatic **CD4-defined** groups, two ART-treated and virally suppressed groups (one with normal CSF NfL and one with elevated CSF NfL, designated as **RxNFL+** and **RxNFL-**) and the asymptomatic escape (**AsE**) group. The HIV-associated dementia (**HAD**), neurosymptomatic escape (**NSE**) and secondary escape (**2ryE**) samples were obtained during clinical evaluations after informed consent. Additionally, the **NSE** and **HAD** groups were augmented by CSF specimens from Milan and Pune to attain a size comparable to the other main groups. When there was a larger number of CSF specimens available for a given group than required for the study, samples were chosen at random from archived collections (e.g., **CD4-defined** groups). An exception to this random selection was the choice of the **HIV-** group from GOT: all of these were from individuals taking pre-exposure prophylaxis (PrEP) because of their self-identified risk for HIV-1 infection. **HIV-** controls from SF were seronegative individuals from the same clinical site and demographic population as the people living with HIV (PLWH) included in the local cohort, but not on PrEP. The **Rx NFL+** and the **AsE** groups were almost all from GOT related to the local interest in these conditions. The small number of secondary escape (**2ryE**) were also all from GOT. In general, the imbalances in the subject group sizes related to scarcity of available samples (e.g., **elites** and **2ryE**, resulting in smaller sample sizes), or to inclusion of larger numbers of specimens to augment particularly important groups, including the **HIV-** group, related to its comparative utility, and **HAD** and **NSE** groups because of our interest in more severe HIV-1-related CNS injury. These imbalances, along with those of calendar years and subject ages (**B** and **C** below), emphasize that this exploratory study used a *convenience sample* rather than one that was balanced with respect to individual demographic variables. **B. Calendar year.** The dates of study varied among the groups. In part this reflected the history of our cohort studies in concert with the changes in acceptance and efficacy of treatment regimens over the collection period. These time variations impacted the length of untreated chronic infection, susceptibility to disease progression and presence of viral suppression among our study cohorts. For example, there were nearly 10 years separating the median years of the **HAD** group from both treatment-suppressed groups (**RxNFL-** and **RxNFL+**). The **NSE** group sampling also reflected later recognition and attention to this clinical entity. **C. Age.** The ages of the subjects were not specifically selected and consequently also varied among groups, with group medians ranging from 39 to 50 years with largely similar ranges. Overall, the median ages of the treated PLWHs were similar to those of the HIV-negative (**HIV-**) controls, while the untreated groups were generally younger. The balance of sex was not taken into account with specimen choices, so distribution also varied among the groups. **D. Blood CD4+ T-lymphocytes.** The median blood CD4+ T-cell count in the **elites** was above that of the uninfected controls and the other HIV-1-infected groups, including the two treatment suppressed groups. The blood CD4+ T-cell concentrations in the 5 groups defined by these counts show the stepwise decrease from >500 to <50 CD4+ T-cells/µL dictated by the study design. The **HAD** group had a median CD4+ T-cell count of 108 cells/µL that was similar to the **50–199 CD4** group median of 106 cells/µL, in keeping with the role of advanced immunosuppression in the development of this subacute disorder and its underlying **HIVE** pathology. The CD4+ T-cell counts of the two treatment-suppressed groups were nearly equal, with both having blood CD4+ T-cell counts above those of the viremic **CD4-defined** groups except for the group with the highest blood CD4+ T-cell counts (**CD4>500** group), indicating the partial CD4+ cell preservation or recovery with suppressive treatment. By contrast, the blood CD4+ T-cell counts in the three CSF escape groups were lower than the two treatment-suppressed groups, consistent with less robust recovery or ongoing loss during the escape episodes. However, these levels were not as low as in the untreated **HAD** group. The CD4+ blood counts of the **2ryE** subjects were also low, indeed below the other two escape groups, likely contributing to their vulnerability to intercurrent infections (herpes zoster in three and HSV2 meningitis in the fourth) and factoring into their clinically-directed LPs. **E. Blood CD8+ T lymphocytes.** Blood CD8+ T-cell counts were on the high side of normal in the **elites**, rose as blood CD4+ cells decreased in the untreated **CD4-defined** groups except for those in the **CD4 <50** group (median of 380 CD8+ T-cells per µL). They were relatively increased in the **HAD** group (median 750 CD8+ T-cells per µL) and in all the treated groups, including the two virally suppressed and the **NSE** and **ASE** groups (medians in 800s per µL). **F.**

**Blood CD4+/CD8+ T-cell ratio.** The differences in the trajectories of the two T-cell subpopulations resulted in lowered ratios in all infected groups compared to the **HIV- controls**, reflecting variable reductions in CD4+ and elevations of CD8+ T-lymphocytes. Reductions were most severe in the **CD4 -defined groups**, most notably in those with CD4+ T-cell counts below 200 cells per µL and in the **HAD** group. These ratios were relatively increased in the four treated groups (0.47 to 0.77), though all remained below the level of the uninfected controls (median ratio of 1.75). Reduced ratios persisted in the two ART-suppressed groups (**RxNFL-** and **RxNFL+**) and the **AsE** and **NSE** groups. **G. Nadir blood CD4+ T-lymphocytes**. Nadir CD4+ T-cell counts are shown only for the ART-treated groups, since CD4+ T-cell values in the untreated groups at their study visits were generally at or near their nadirs. The treated groups showed varying degrees of presumed blood CD4+ T-cell recovery, higher in the two virally suppressed (**Rx NFL-** and **NFL+**) groups (medians of 590 and 600 CD4+ T-cells per µL) than in the CSF escape **NSE** and **ASE** groups (medians of 425 and 412 CD4+ T-cells per µL, respectively), the latter perhaps either contributing to development of CSF escape or, alternatively, reflecting an impact of escape on CD4+ T-cell dynamics. **H. CSF HIV-1 RNA**. Highest CSF HIV-1 RNA concentrations were present in the **HAD** group, while the **CD4-defined** groups showed the 'inverted U', or *lymphoid*, pattern of change with lower concentrations at the extremes (in the **CD4+ >500** and **CD4+ <50** groups) than in the middle ranges (**CD4+ 350–499**, **200–345** and **50–199** groups). We previously reported the association of this pattern with that of the CSF WBC counts as also noted in this study (see **Fig 1K** below), suggesting that these two findings were causally related [45]. In this study the correlation of CSF HIV-1 RNA to CSF WBC count across the **CD4-defined** groups was significant (Pearson correlation P = 0.006). This lymphoid pattern of infection and cell response likely underlies this same pattern in many of the CSF proteins included in the **Olink Explore 1536** panel as shown in later figures. The CSF HIV-1 RNA concentrations of three escape groups were generally below those of the untreated groups except for the **CD4 <50** group. CSF HIV-1 RNA concentrations were all below detection in the **elites**. **I. Blood HIV-1 RNA**. In the untreated individuals (not including the **elites**) there was a steady increase in blood HIV-1 RNA through the full range of CD4+ T-cell loss that reached its highest mean levels in the **HAD** group. Unlike with CSF, there was no decrement in the **CD4 <50** group, showing a distinct difference between the CSF and the systemic blood viral dynamics. The presence of low-level blood HIV-1 RNA in the **NSE** group may have reflected systemic partial drug resistance in some or spillover of the 'escaped' CNS/CSF infection into the blood. **J. CSF:Blood HIV-1 RNA differences**. This panel shows the differences in viral loads between the two fluids (calculated as the $\log_{10}$ CSF HIV-1 RNA copies per mL–$\log_{10}$ plasma HIV-1 RNA copies per mL) and emphasizes the largely consistent relationship between CSF and blood HIV-1 RNA levels in untreated infection at blood CD4+ T-cell levels between 50 and >500 cells/µL in which the CSF concentrations were approximately 10-fold lower than those in blood, and thus maintaining a nearly 1:10 ratio of CSF to blood HIV-1 RNA. This may relate to the kinetics in the influx of infected CD4+ T-cells and of viral release into the CSF over this blood CD4+ T-cell range. This ratio was disrupted when blood CD4+ T-cells fell to <50 cells/µL and the CSF WBC count decreased to negligible levels; in this group the CSF:blood HIV-1 RNA ratio decreased nearly 10-fold to an overall ratio of <1:100. This underscores the importance of lymphocyte traffic and likely direct virus release by trafficking infected CD4+ T-cells in the determining the CSF HIV-1 levels in the neurologically asymptomatic individuals, particularly those with blood CD4+ T-cell counts above 50 cells per µL. This relationship changed markedly with the development of **HAD** and the underlying direct neuropathic parenchymal CNS infection, i.e., **HIVE**, in which the CSF HIV-1 RNA concentrations reached high levels, and the differences between it and the blood viral load decreased. Thus, in the **HAD** group local brain infection rather than hematogenous sources likely was responsible for the high levels of CSF HIV-1 RNA. The reversed CSF:blood HIV-1 RNA ratios in the three CSF escape groups were consonant with their definitions predicated on CSF > blood HIV-1 RNA concentrations that indicated the direct CNS sources of the measured CSF HIV-1 RNA in the face of systemic viral suppression. **K. CSF WBC counts**. The median CSF cell counts rose and then fell over the course of untreated infection in the **CD4-defined** groups. CSF WBC counts were highest in the 200–349 group and declined to lowest levels when blood CD4+ T-cells fell below 50 cells/µL, defining the *lymphoid* pattern. As discussed above, this likely was causally linked to the changes in CSF HIV-1 RNA in these groups. Both neurologically defined groups were associated with augmented local inflammation: CSF WBCs were elevated in **HAD**, and even higher in **NSE** (median counts of 10 and 22 cells per µL, respectively). In these two settings the CSF WBC counts likely involved a response to local CNS infection and injury. **L. CSF neopterin**. This pteridine is predominantly, though perhaps not exclusively, a macrophage-related activation marker [143]. CSF levels showed a steady median increase as CD4 cells declined, including the highest levels in the **CD4+ <50** group among the **CD4-defined** sequence of groups, thus providing a prototypical example of the *myeloid* pattern of CSF biomarker change. CSF concentrations were even higher in the **HAD** group, and, notably, were highest in the **NSE** group, while suppressive treatment brought CSF neopterin concentrations back to near normal levels in the absence of symptomatic CSF escape. **M. Blood neopterin**. In blood, neopterin concentrations showed a similar, though more restricted elevation with CD4 decline but were notably increased in both the **CD4 < 50** and **HAD** groups, presumably indicating systemic macrophage activation in these settings that either links these two sites of infection or marks parallel changes in myeloid cell populations in the CNS and systemically. In contrast to CSF, the blood neopterin was only mildly elevated in the **NSE** and **2ryE** groups in keeping with the underlying compartmentalized CNS HIV-1 and intercurrent infections in these two groups in the face of systemic HIV-1 suppression. **N. CSF: blood albumin ratio**. This ratio provides an index of blood-brain barrier integrity, though with considerable individual and age-related variability [144]. In this sample set, only direct **HIVE** in the **HAD** group and, to a lesser degree, in the **NSE** group was associated with elevated median albumin ratios. Minor, though not significant, increases were present in the **RxNFL+** group. One-way ANOVA with Tukey's multiple comparison test found significant differences (P<0.05) only for **HAD** (CD4 >500: P = 0.0004; Elites: P = 0.0028; AsE: P = 0.0067; CD4 200–349: P = 0.0069; RxNFL-: P = 0.0116; CD4 50–199: P = 0.0202; CD4 350–499 P = 0.0228) and **NSE** (CD4 >500: P = 0.0023; Elites: P = 0.0075; AsE: P = 0.0173; CD4 200–349 P = 0.0211; HIV-1- P = 0.024; RxNFL- P = 0.0337; CD4 50–99 P = 0.0479). Other intergroup comparisons were not significant. Overall, the albumin ratio elevations were small, and likely to have had only limited, if any, impact on the CSF proteins measured in this panel. **O. CSF NfL.** Prior to this study, CSF NfL was measured using the UMAN ELISA method (UmanDiagnostics, Umeå, Sweden) in 209 of the 307 (71.3 percent) of the specimens. In this figure, the NfL values have been adjusted to age 50 years using the normative data and methods of Yilmaz and colleagues [68] to allow more direct subject comparisons and estimations of abnormal values. While these data were incomplete and superseded in this study by the Olink measurements performed on all samples, they guided our initial definition of the **RxNFL+** group. The highest levels of NfL were in the **HAD** group, and there were elevated levels in a substantial proportion of the **CD4 <50 group** presumably related to subclinical neurological injury. Notably, the NfL elevation of the **HAD** group had nearly a 10-fold higher median value than that of the **NSE** group. These observations were confirmed and extended by the measurement of NfL within the Olink Explore panel as shown later in S2 **Fig**.

untreated groups with higher blood CD4+ T-cell counts [17]. The low CSF WBC counts in the **CD4 <50** group may have related to the limited availability of both CD4+ and CD8+ T-cells that generally comprise the majority of CSF WBCs in HIV-1 infection [83]. More particularly, the low number of CD4+ T-cells in this group likely resulted in decreased passage of HIV-1-infected cells from the blood into the leptomeninges, and, consequently, diminished viral

release into the CSF spaces. The reduced presence of CSF T-cells in this group also likely impacted the levels of some of the measured proteins in this study, resulting in this same *lymphoid* pattern for many of the CSF proteins in the sequence of **CD4-defined** groups, similar to the previously reported patterns in a limited subset of inflammatory biomarkers that included CXCL10, TNF-alpha and MMP-9 as discussed above in the **Introduction**. In contrast, the CSF HIV-1 RNA concentrations in **HAD** group increased to the highest levels among the groups, to near those in the blood (CSF median of 5.27 and blood median of 5.66 $\log_{10}$ copies RNA per mL). This was a consequence of the **HIVE** underlying **HAD** in which local CNS HIV-1 replication within the brain likely 'spilled over' into the CSF [17]. The median CSF WBC count in the **HAD** group was 10 cells per μL, and the median WBC count was even higher in the **NSE** group (22 cells per μL), though in the latter group the virological impact of the pleocytosis was likely mitigated by a partial treatment effect. The higher CSF than blood HIV-1 RNA concentrations in the three CSF escape groups (on therapy with virus suppressed in the blood) defines these entities, with the **AsE** group's CSF virus concentrations (median of 1.90 $\log_{10}$ copies per mL) more than 10-fold lower than the other two escape groups (median CSF RNA concentration in the **NSE** group of 3.23 and in the **2ryE** of 2.94 $\log_{10}$ copies per mL). Though it presents an interesting comparison to **NSE**, the **2ryE** group was small and included CSF from three individuals with herpes zoster and one with HIV-2 meningitis, so it cannot be considered as representative of the larger possibilities in this category.

The virally suppressed (**RxNFL-**) group was likewise typical of the effectiveness of ART in normalizing symptomatic or asymptomatic neural injury. By contrast, the **RxNFL+** group was selected from our larger experience based on elevated age-adjusted CSF NfL measurement in the absence of neurological symptoms and signs. The causes of these elevations in NfL were unknown and did not appear to predict neurological deterioration in these individuals during available follow up. Among the main possibilities are that they might have reflected mild brain insults unrelated to HIV infection (e.g., trauma) or that they were due to transient subclinical HIV-1-related injury. The lack of association with concurrent elevations in CSF HIV-1 levels, or with changes in the other CSF neural or inflammatory biomarkers (as described below) likely argues against the latter and suggests they might not be related to CNS HIV-1 infection. We plan to examine these groups further in the future to address these issues.

## CSF Neopterin: <u>Myeloid</u> pattern of change in the CD4-defined groups

Neopterin is a pteridine inflammatory mediator that is likely mainly produced by myeloid and possibly to a lesser extent by astroglial cells within the CNS [145–147]. It serves as a useful CSF inflammatory biomarker during HIV-1 infection [75]. In contrast to HIV-1 RNA and WBCs, CSF neopterin showed a steady increase with falling CD4+ T-cell counts in the untreated PLWH groups, without a subsequent decrease in the **CD4 <50 group** (**Fig 1L**). Based on this and previous observations with other myeloid-related biomarkers, including CCL2 and sCD163 [45], we have now termed this the *myeloid* pattern of biomarker change within the untreated groups that contrasts with the *lymphoid* pattern discussed above. We use these two convenient, albeit simplistic, terms in this report to designate recurring contrasting patterns of biomarker changes observed in a number of the Olink-measured CSF proteins across the CD4-defined groups as presented below. The **HAD** group and two of the three escape groups, **NSE** and **2ryE**, exhibited marked elevations in CSF neopterin while median concentrations in the **AsE** group were near normal. Blood neopterin levels also rose in the untreated individuals; they were notably elevated in the CD4 < 50 group and also in the HAD group, consistent with systemic myeloid activation in these settings [28]. By contrast, blood neopterin levels in the three escape groups remained at or near normal despite the elevations in CSF, including in the

*NSE* and *2ryE* groups with notably high CSF values, consistent with the compartmentalized CNS infection and inflammatory responses in these groups and local production of neopterin. In *AsE* the median neopterin values in both fluids were at or near normal with the exceptions of three CSF samples, distinguishing these individuals from the other two escape groups (*NSE* and *2ry*) and suggesting that *AsE* is a transient and less pathogenic form of CSF escape [98]. This contrasted with the parallel CSF and blood neopterin changes with CD4+ T-cell decline in the untreated individuals and with the generally low blood neopterin levels in the virally suppressed (*RxNFL-* and *RxNFL+*) groups [45].

## CSF NfL in a subset of the CSF sample

**Fig 1O** shows the results of the CSF NfL measurements that had been assayed in a subset of the samples at the University of Gothenburg Clinical Neurochemistry Laboratory over a number of years in several different analytical runs of an enzyme-linked immunosorbent assay (ELISA) and age-adjusted to 50 years of age using a previously described method [68]. There were consistent CSF NfL elevations in the *HAD* and *NSE* groups (more than ten-fold higher in the former) and milder increases in about half of the *CD4 <50* group, consistent with relatively common subclinical CNS injury in the setting of advanced immunosuppression. The general pattern of NfL change across this group showed similarity to the *myeloid* pattern discussed above, but a few possible differences can be noted: the elevation of NfL in the CD4 <50 group was 'abrupt' without a clear lead-up in the other CD4-defined groups; additionally, there was an increase in CSF NfL in the *RxNFL+* group that was absent in CSF myeloid or other neural markers as discussed below. More generally, the similarities between the patterns of change of the CSF *myeloid* markers and CSF NfL, that defines a *neuronal* injury pattern in CSF, likely relates to the importance of myeloid cells in the pathogenesis of CNS injury.

## Introduction to the Olink Explore output and CSF dataset with analysis of repeated measurements

Proteins measured using the **Olink** platform are reported in NPX units on a $\log_2$ scale of *relative concentrations (https://olink.com/products/olink-explore-3072-384)*. Thus, the output allows comparison of the measured protein concentrations of individuals or groups *only within this particular study*, and would require simultaneous calibration, with assessments of known concentrations for conversion to more standard quantitative units (e.g., ng/mL). Consequently, interpretations of results for the HIV-1-infected groups depends importantly on comparisons with the **HIV-uninfected controls** measured within this same assay.

The Olink output measuring 1,472 proteins includes quadruplicate measurements of three proteins (CXCL7, IL6 and TNF) as outlined below. To eliminate the effect of this redundancy, we randomly selected only one set of results for each of these three proteins to include in our overall analysis. This reduced the analysis to include assays of 1,463 unique proteins applied to 303 specimens, yielding a total of 443,289 individual CSF protein measurements. However, some of the measurements fell below the LOD of the individual assays. As noted above in the **MATERIALS AND METHODS** section, in this report we used the measured values without any correction related to the LODs, since these low values likely had little more effect on analysis than would truncating or eliminating values below the LODs. For other investigators who may want to adjust these values differently in future studies, the four LOD values for each protein assay are included in the posting of the full data set (File 1 Dataset of CSF proteins in chronic HIV _infection_rev2024-09-10.xlsx (DOI: 10.5061/dryad.x3ffbg7tv)) [134].

### Analysis of three sets of quadruplicate repeat measurements; features of Olink Explore data output

As a prelude to the main analysis, we examined the intercorrelations among the quadruplicate measurements of three CSF proteins: CXCL8, IL6 and TNF. The Olink Explore 1536 platform includes four independent measurements of these proteins, one on each of the four assay plates. Within these data we also examined the impact of Quality Control (QC) Warnings that flagged some of the measurements and the frequency of protein measurements below the LODs reported in the Olink data output file. The QC Warnings and the percentage of measurements within different strata of LODs (100; 90 - <100; 60 - <90; 40 - <60; 20 - <40; and <20 percent) are all indicated within the full dataset table (File 1 Dataset of CSF proteins in chronic HIV _infection_rev2024-09-10.xlsx (DOI: 10.5061/dryad.x3ffbg7tv)) [134].

The analyses outlined in **S1A-S1C Fig** show the high intercorrelations among the quadruplicate NPX values for two of the three sets of quadruplicate values and lower correlations in the third (mean $R^2$ results of the 6 pairwise comparisons for each of the proteins were 0.9821 and 0.9508 for CXCL8 and IL6, with less robust correlation for the TNF correlations with a mean $R^2$ = 0.7055). These correlations, along with very similar patterns of concentration changes across the subject groups (**S1B Fig**), were present despite differences in the absolute NPX values in some of the repeats, emphasizing the *relative scale* of the Olink protein measurements and the importance of a full set of embedded controls in each independent assay.

In all but two cases the measurements with the QC warnings in these assays fell within the confidence limits of the overall results. Based on this finding, and to reduce biased selection of the Olink output, we included all Olink results in the analyses that follow, including those with QC Warnings. Values below the LODs were minimal in the CXCL8 and IL6 repeated measures and did not impact correlations. However, a greater frequency of results below the LODs were noted with the TNF measurements and contributed to less robust correlations among the four repeated measures of this protein, though perhaps not providing a complete explanation. As noted above, for the analyses in this report, we did not exclude assay results below the LODs that are listed in the Dryad posted (File 1 Dataset of CSF proteins in chronic HIV _infection_-rev2024-09-10.xlsx (DOI: 10.5061/dryad.x3ffbg7tv)) and discussed in the legend to **S1 Fig**.

Values below the LODs preferentially affected the analysis of the aviremic groups (e.g. ***HIV-***, ***elites***, ***Rx NFL-*** and ***NFL+***), reducing the potential of some proteins to discriminate differences among these groups. Because of attention to the overall protein changes, and particularly to the differences among the ***CD4-defined*** groups and the ***HAD*** and ***NSE*** groups, this issue likely had relatively limited impact on the main analyses included in this report. However, it likely will be more important in future studies using these data to examine some of the proteins in these aviremic groups in which the values below the LODs will assume greater prominence.

### Olink CNS injury markers and the use of the Olink measurement of NEFL as a key biomarker

In an additional preliminary study, we examined the Olink measurement of NfL, that is included in the **Explore 1536** platform, along with six other biomarkers of CNS injury in the panel. In several past studies we have shown that measurement of NfL in CSF (as well as in blood) serves as a valuable biomarker of active CNS injury in HIV-1 infection [66,68], just as it does in head injury, subarachnoid hemorrhage and a number of neurodegenerative conditions [65,148]. Because prior CSF NfL measurements were only available from a portion of the CSF specimens included in this study (209 of the 307, 71.3 percent), we evaluated whether the Olink measurements which are available for all specimens would be suitable for use as

*background variable* in the main analyses of this study. [**NB**: In this report, we use the gene name, *NEFL*, when referring to the Olink measurement of the protein, and the more common abbreviation, *NfL*, when referring to the protein more generally and to results of measurements using other, conventional immunoassays.]

S2**A Fig** shows that the profile of Olink measurement of NEFL changes across the subject groups is very similar to that obtained with the NfL ELISA immunoassay performed in the subset of the sample shown earlier (comparing **S2A Fig** to **Fig 1O**). In fact, the results of the two assays are highly correlated ($R^2 = 0.8790$) (S2**B Fig**). These comparisons thus support the use of the NEFL results as a principal objective marker of CNS injury in the subsequent analyses. We do not age-adjust the NEFL values in this analysis because we have not age-adjusted any of the other Olink protein results and do not have sufficient data to validate age adjustment for the Olink NEFL results. The mean NEFL values in the *RxNFL+* group were higher than the *RxNFL-* group, indicating that the earlier immunoassay measurements in the former group were not simply laboratory errors but reflected true elevations of the CSF concentration and likely genuine axonal injury, albeit of uncertain source and meaning.

The **Olink Explore 1536** platform includes several other markers that have been used in previous studies to assess CNS injury in HIV-1 infection and other neurodegenerative settings (S2**C**–S2**H Fig**). Of these, CSF CHI3L/YKL-40, TREM2, GFAP, KYNU and MAPT (all P $<0.0001$ with $R^2$ = 0.6005, 0.5622, 0.4545, 0.3690 and0.3471, respectively) warrant further study as useful complements to NEFL in assessing the breadth and character of CNS injury as discussed in the legend to this figure. Of note, the elevation of CSF NEFL in the *RxNFL+* group was not echoed by elevations of these other neural injury markers, adding further uncertainty to the biological meaning of the CSF concentration increases of this axonal protein in this group.

## Main Olink CSF protein analyses

The following three main analyses explore facets of the full dataset of the measured CSF proteins as they varied over the course of systemic infection, development of HAD, successful treatment and CSF escape as represented by the specimen groups assembled for this study. The first of these analyses provides a broad introduction to the overall CSF protein changes across the study groups. The second examines the relationships of these protein changes to two major variables of CNS HIV-1 infection and its impact: CSF HIV-1 RNA concentration (as an index of CNS infection) and CSF NEFL concentration (as a measure of active CNS injury). The third compares CSF proteins in two forms of *HIVE*: *HAD* presenting in untreated individuals and *NSE* developing during treatment.

### Overview of the CSF protein changes across the spectrum of chronic HIV-1 infection

As an initial step in analyzing the broad contours of the protein changes during the course of chronic infection, we performed a hierarchical cluster analysis that empirically groups the measured proteins into three clusters with different spectra of protein changes across the full sample set. These are shown in **Fig 2A** as the **BLUE**, **RED** and **GREEN** clusters.

The initial division into three clusters in this figure provides a starting point for identifying proteins and groups of proteins with similar or contrasting overall protein profiles. As detailed in the legend to **Fig 2**, the widest and most distinct gradations in protein concentrations are grouped in the **GREEN** cluster with the conspicuously high protein concentrations (at the red and orange end of the scale) in the *HAD*, *NSE*, and *2ryE* groups. These contrast with the lower concentrations (in the blue or blue mixed with yellow bands) in the aviremic *HIV-*, *elites*, and

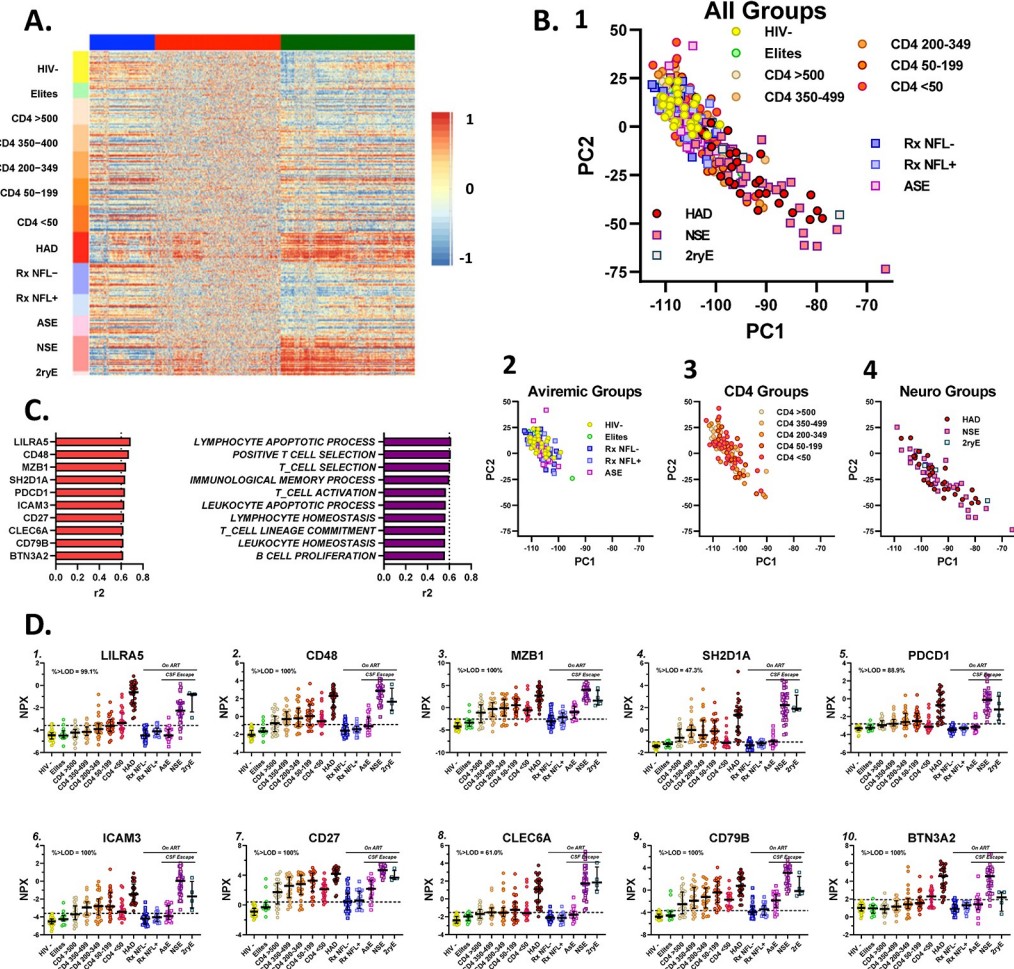

**Fig 2. Overview of CSF proteins across subject groups.** The assembled panels depict some of the main features of the CSF protein changes across the set of subject groups. **A.** This ***hierarchical cluster analysis*** displays the full set of the Olink CSF protein measurements in a heatmap. The study groups (and the individual CSF samples within each group) are arranged on the vertical axis, while the proteins are grouped across the figure after segregation into three clusters designated by colors at the top within **BLUE**, **RED** and **GREEN** columns of the measurements of individuals proteins in each sample. The heatmap relative scale from -1 (blue) to 0 (yellow) to 1 (red) is shown to the right of the main figure. The individual proteins are color-scaled according to their relative concentrations. The most conspicuous gradations in scale are within the **GREEN** cluster (right column) in which many samples from the ***HAD*** group and the two symptomatic escape groups (***NSE*** and ***2ryE***) stand out with higher (red) concentrations compared to the generally lower concentrations (blue and yellow) in the non-viremic individuals (***HIV-***, ***elites***, and ***RxNFL+*** and ***RxNFL-***) with some mixed colors over this spectrum in the ***CD4-defined*** groups. By contrast, the **RED** cluster (middle column) showed a more even and mottled spectrum except for some of the proteins shown on the left side of this cluster in which there was red predominance in the ***HAD*** and two symptomatic escape groups similar to, but less pronounced than, in the **GREEN** cluster. The changes among groups in the **BLUE** cluster (left column) are less distinct but with some low (blue) concentrations in the ***CD4 <50***, ***HAD*** and two symptomatic escape groups (***NSE*** and ***2ryE***), and higher concentrations in the HIV- group, i.e., showing some patterns that are *opposite to* (or inverted in comparison to) those in the **GREEN** cluster. Overall, the ***HAD*** and two symptomatic escape groups accounted for the greatest concentration differences from the controls among the proteins, most prominently in the **GREEN** cluster. **B.** This ***PCA analysis*** using results of all the measured CSF proteins to assess effects of the overall CSF measurements on individual subjects and their groups provides a different perspective on the protein changes across the groups. **1.** This panel includes the results from all the groups and shows a 'gradient' downward and to the right of the panel associated with HIV-1 systemic disease progression (as indicated by falling blood CD4+ T-cell counts) and, more particularly, with the symptomatic neurological disease groups, so that individuals with ***HAD*** and ***NSE*** reach farther in this direction. Because of the superimposition of many of the individual in this panel, three smaller panels (**Fig 2B2-4**) separate these groups for better visual definition of these changes. **2.** the aviremic groups including the ***HIV-***, the largely superimposed ***elites***, treated-suppressed (***RxNFL-*** and ***RxNFL+***) and the ***ASE*** group are all grouped together with overlap in the upper left of the plot; **3.** the five ***CD4-defined*** groups are also largely clustered in the same upper left area but with some individuals 'moving'

down and to the right as immunosuppression advanced with lower blood CD4+ T-cell counts; *4.* the **HAD** and two symptomatic escape groups (**NSE** and **2ryE**) showed some overlap with the other groups but with further movement down and to the right (higher PC1 and lower PC2 values), again consistent with these neurological groups importantly 'driving' the main group differences in the aggregate CSF protein changes across the large cohort sample. Thus, these groups, defined clinically, are also, not surprisingly, separated in this analysis by the changes in their CSF proteomes. **C.** P*roteins and pathways influential in distinguishing subject groups*. The two bar plots show the proteins and pathways that most influenced the overall differences among the subject groups. The left bar graph lists the most influential proteins, all 10 of which are markers of inflammatory processes. The right bar graph lists the most influential defined pathways extracted from the protein changes, and again their designations emphasize that they identify inflammatory processes, including several involving T-cells. While, in part this may reflect the large number of inflammatory proteins that were included in the **Explore 1536** panel, many with intersecting or overlapping functions, it also clearly shows that changes in inflammatory profiles largely determine the contours of the overall measured CSF proteome as systemic disease progressed and, particularly, as overt CNS disease with **HIVE** developed. **D.** *Influential protein patterns across study subjects*. These plots show the subject group concentrations of the 10 most influential proteins (from the left side of **Fig 2C** above) across the subject groups using the color schema introduced in **Fig 1** and used throughout this report. This includes the color symbols, medians and IQR bars, and the dashed horizontal line that indicates the mean + 2 SD of the **HIV- control** group in this study as an estimate of the upper limit of these variables and aids as visual reference across the groups. Protein concentrations are expressed in $\log_2$ NPX units with varying scales. The plots show the prominence of the increases in the **HAD** and **NSE** groups and their dominating influence; these two groups had the highest concentrations of all 10 of these CSF proteins, with lower and more variable patterns of elevation in the CD4-defined and other groups. As annotated in the panels, of the 10 listed proteins, the measurements in six were all (100%) above the LODs for the assay. The exceptions included LILRA5, SH2D1A, PCDC1, and CLEC6A in which 99.1%, 47.3%, 88.9% and 61.0%, respectively, of the proteins were above the LODs (shown in the individual panels here as in subsequent figures of protein patterns across the subject groups). These influential proteins are all within the **GREEN** cluster in **Fig 2A** and are listed in the *Glossary of Featured Proteins in Fig 2* included in **S1 Appendix** with brief functional descriptions extracted from UniProt database (https://www.uniprot.org/uniprotkb?query=*) [140], along with comments on selected features of the individual panels.

treatment suppressed (**RxNFL-** and **RxNFL+**) groups, and the more heterogeneously colored bands in the **CD4+-defined** groups. Similar, though less distinct and less consistent, gradations are also present on the left side of the **RED** cluster column, while the remaining two-thirds of the **RED** group show more even concentrations (and thus narrower concentration differences) through the full range of specimens. While some of these were likely proteins that were not altered over the course of infection, others are proteins with concentrations outside of the levels of measurement of the Olink assay, mainly below the LODs (see below). The **BLUE** cluster shows more mixed gradations, including a substantial number of proteins with seemingly 'inverted' concentrations—i.e. with lower concentrations in the **HAD** and **NSE** groups and in the lower **CD4 defined** groups than in the **HIV- controls**. These initial similarities and contrasts bear further examination within smaller groupings and subdivisions of these clusters in the future.

The principal component analysis (PCA) (**Fig 2B1–2B4**) shows the impact of the overall CSF protein changes on separating the individual participants and their groups. The virologically undetectable PLWH (**Elites**, **Rx NFL-** and **NFL+**, and **ASE** groups), along with the **HIV-** control group, all largely overlapped in this figure. The **CD4-defined** groups also overlapped with these same groups, but with some individuals with lower CD4 counts 'moving down and to the right' in the compound graph and again seen more clearly in isolation (**Fig 2B3**). Most notably, the **HAD**, **NSE** and **2ryE** groups 'migrated' still further in this same direction with greater separation of many of the samples from the other groups (**Fig 2B4**) consonant with the higher concentrations of many CSF proteins in these groups. Given the conspicuous elevations of many proteins in the HAD, NSE and 2ryE groups, one might ask why the separation of subjects in this PCA is not greater. In part this may relate to inclusion of *all measured proteins* in the analysis, including those that show little (**RED** proteins in **Fig 2A**) or even opposite protein gradients (some of **BLUE** proteins in **Fig 2A**), that might 'counterbalance' or obscure the distinct group differences and reduce the separation exerted by the proteins in the **GREEN** group in **Fig 2A**).

Together the cluster analysis heat map and the PCA provide two consonant perspectives on the same CSF protein changes: the groups identified in the heat map as having the highest concentrations of many CSF proteins (the **HAD**, **NSE** and **2ryE** groups) are segregated furthest from the **HIV-** uninfected and neurologically asymptomatic groups in the PCA analysis. Thus, these symptomatic neurological conditions have a dominating effect on the CSF concentrations of many proteins that, in turn, results in segregation of these neurological subject groups from those without neurological disease based on empirical CSF protein analysis.

**Fig 2C** lists the 10 proteins and 10 pathways that exhibited the most significant differences in CSF proteins across groups. Most of these are inflammation-related proteins, and the defined pathways involve lymphocyte-related functions as their names imply. These proteins are associated with T-cell, B cell and NK cell functions. Thus, although the clinical settings of two types of neurological injury had the greatest impact on the separation of the proteins, it was the inflammatory responses in CSF in these settings that dominated their impact among the measured proteins. **Fig 2D** shows the group profiles of the concentration changes across the subject groups of the 10 highest correlating proteins listed in **Fig 2C**. While the overall patterns varied, the most conspicuous common feature is the high protein levels in the **HAD** and **NSE** groups (highest for the former in some, but in this group of proteins more frequently highest in the latter). These proteins are all from the **GREEN** group in the cluster analysis in **Fig 2A**, and their profiles again emphasize that the two neurological disease groups are the important drivers of the greatest CSF protein differences in the sample set. These graphs also introduce other prominent features of the CSF protein spectra across subject-group, including the two patterns of CSF biomarker change over the course of CD4+ T-cell decline in the five untreated **CD4-defined** groups introduced earlier:

i. The *lymphoid* type, with highest levels in the **CD4 50–199** or **200–350** groups set off by lower levels in the **CD4 >500** and **CD4 <50** groups (and most notably the latter) in the 'inverted U' pattern discussed earlier in the context of the HIV-1 RNA and WBC concentrations. This is the most frequent pattern in this group of proteins.

ii. the *myeloid type*, with progressive increases leading to highest levels in the **CD4 <50** group as noted earlier with CSF neopterin, is less frequent but exemplified here with the LILRA5 and BTN1A2 proteins at the upper left and lower right of this panel.

Although these ten proteins provide examples of the highest correlations, the PCA segregation of the subject group involves the entire set of measured proteins, including not only those that determined the groups separations but also those that 'pulled' them together because of limited protein concentrations differences. Also, while a number of these proteins share features with the *lymphoid* profile and two with the *myeloid* profile, there are also variations within these individual profiles that bear focused and more nuanced analysis in the future.

These data also illustrate the limited view provided by most previous studies, including our own, that measured only a few selected inflammatory mediators [2,45,75,82,149]. The results clearly underscore both the complexity and the broad participation of many proteins, including particularly many inflammatory proteins, in response to CSF/CNS infection and CNS injury. The findings, of course, do not distinguish which proteins *contribute* to CNS injury rather than simply *respond* to it. Nor do they distinguish which inflammatory proteins are involved in initiating and sustaining or determining the magnitude and character of the group inflammatory profile. These, and other, crucial mechanistic issues must be inferred and synthesized from the known functions of the proteins and their relative responses in the different clinical groups. In this report we focus on the empirical findings and leave a more fully synthesized mechanistic construction to future extensions of these analyses.

Of note, at least three of these proteins show mild elevation in the *AsE* group, indicating the low-level inflammatory responses in this setting that warrant future focused attention to this group. Further, while the *HIV-*, aviremic *elites* and the two treated groups (*RxNFL-* and *RxNFL +*) overlap in this analysis, it will be of interest to examine these groups further to identify possible distinctions among these groups to determine to what extent and how endogenously controlled (in *elites*) and therapeutically controlled infections may differ. We have not included these and other analyses in this introductory report, but plan to pursue these and other studies in the future.

## CSF protein relations to CNS injury (CSF NEFL) and infection (CSF HIV-1 RNA)

As a step in dissecting the forces underlying the changes in CSF proteins across the subject groups, we examined the relations of CSF proteins to two major variables that can be viewed as indicators of key *pathogenetic vectors* involved in these changes: **1**. CSF HIV-1 RNA concentrations as an *index of CNS HIV-1 exposure/infection* (encompassing entry, production and replication, with the latter two likely the main presumed drivers of CNS injury when located within the CNS parenchyma), and **2**. CSF NEFL concentrations as a measure of the state of *active CNS injury* at the time of sampling. CSF HIV-1 RNA is an ambiguous indicator of CNS infection because it can reflect a variable mixture of virions derived from leptomeningeal and brain parenchymal sites of infection, depending on the clinical setting. During neuroasymptomatic infection in the *CD4-defined* groups (particularly those with CD4 T-cell counts >50 cells per μL), infection is largely meningeal, while *HAD* and *NSE* involve deeper parenchymal infection (*HIVE*). However, the contributions from these sources of origin cannot be distinguished by simple measurement of HIV-1 RNA in CSF. In **Fig 3** we examine the relationships between the various CSF proteins and CSF NEFL and CSF HIV-1 RNA concentrations.

As shown in this figure, CSF NEFL and HIV-1 RNA concentrations are, themselves, not strongly intercorrelated across the full spectrum of study subjects (**Fig 3A**). Even after segregation of subsets of the groups (untreated and treated groups; and *HAD* and *NSE* groups in isolation), correlations between CSF HIV-1 RNA and NEFL remained weak. **Fig 3B** shows the 10 proteins that separately correlated best with CSF HIV, CSF NEFL, and with both of these variables taken together. For a broad overview of the protein relations to CNS injury and infection, we plotted all of the Olink proteins in relation to their correlations with CSF NEFL and HIV-1 RNA (**Fig 3C**). Proteins from the **GREEN** cluster dominate the higher correlations with both NEFL and HIV RNA, and a higher proportion of the proteins correlate with NEFL than with HIV-1 RNA at levels above 0.5 (designated by horizontal and vertical dotted lines). This is consonant with the protein gradients noted earlier for the **GREEN** cluster showing highest concentrations in the *HAD* and *NSE* subjects and the dominant effect of these two overt neurological conditions on elevated CSF protein concentrations.

By contrast, proteins from the **RED** cluster are mainly located in the center of **Fig 3C**, with HIV-1 RNA correlation values centering around zero and relatively low NEFL correlations (below 0.5) except for a few higher outliers. Many of the **RED** proteins returned >60% of measured values below their LODs, likely contributing to their frequent lack of correlation with either HIV-1 RNA or NEFL. Interestingly, the **BLUE** cluster is dominated by proteins with negative HIV-1 RNA correlations and NEFL correlations below the 0.5 correlation level, though with several showing negative correlations with CSF HIV RNA despite >0.5 correlation with NEFL, mingling with some of the **GREEN** cluster proteins in this location. This figure also helps to identify CSF proteins with similar functional properties (or at least correlations) and can provide a useful initial map for future studies exploring functional relationships among the CSF proteins in HIV-1-related neuropathogenesis.

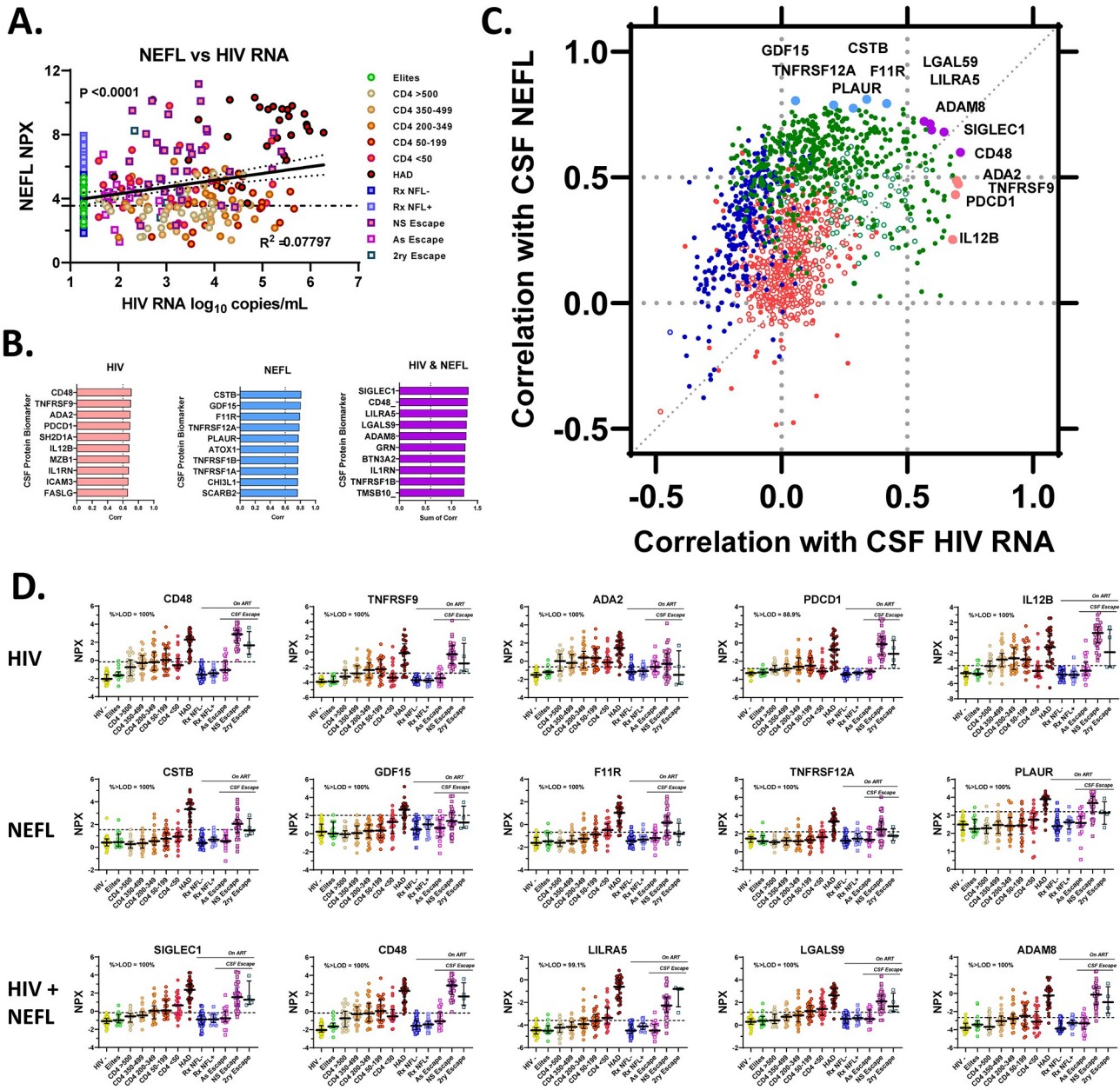

**Fig 3. CSF proteins in relation to correlations of CSF NEFL and HIV-1.** This figure explores the relationships of the CSF proteins to two major variables (or *pathogenic vectors*) of HIV-1-related CNS disease: CSF HIV-1 RNA, an ambiguous index of CNS infection, and CSF NEFL (NfL measured by Olink in the ***Explore 1536*** panel), an objective measurement of active CNS injury. **A**. **Correlation of NEFL with CSF HIV-1 RNA**. This plot shows a significant (P<0.0001), but relatively weak correlation ($R^2$ = 0. 0.07797) between NEFL and CSF HIV-1 RNA by linear regression. Among the features underlying this limited correlation are the high CSF RNA values without elevation of NEFL in the untreated CD4-defined groups, the elevations of CSF NEFL in both the treated ***NSE*** group despite their relatively lower CSF HIV-1 RNA levels (related to the partial suppression from ART) and the untreated ***HAD*** patients with high levels of HIV-1 RNA. Separate analysis of the untreated and treated groups showed little improvement of the correlations (both also with P< 0.0001, but with $R^2$ values of 0.1699 and 0.1373, respectively). Analysis of HAD patients alone showed p = 0.1709 ($R^2$ = 0.07087). **B**. **High correlating individual proteins**. The three bar graphs show the 10 CSF proteins with: the highest correlations to **1**. CSF HIV-1 and **2**. CSF NEFL; and **3**. the sum of the CSF HIV-1 and CSF NFL correlations. For 27 of the listed proteins, the measurements are above the LODs. Exceptions include SH2D1A in which only 47.3% of the proteins are above the LODs, and consequently this protein is omitted from further descriptions and the plotting of the proteins across subject groups below. The other two exceptions are PCDC1 (88.89% above LODs) and LILRA5 (99.1 above the LODs) which are included in the further discussion and illustration. Notably some of these proteins were also previously identified as strongly impacting the overall segregation across groups as defined earlier in **Fig 2**. **C**. **Correlations of the full set of CSF proteins with CSF NEFL and HIV-1 RNA**. This figure includes a total of 1662 proteins, i.e., all of the Olink-measured proteins except NEFL, plotted according to their correlations with CSF NEFL and CSF HIV-1 RNA. It provides a broad overview of the measured proteins in the context of these two important pathogenetic vectors. Each protein symbol is colored according to their grouping in the three clusters (**GREEN**, **BLUE** and **RED**) defined earlier in the hierarchical cluster analysis (**Fig 2A**)

with the solid symbols showing the proteins with ≥60% of the proteins above their LODs and the open symbols showing the proteins with <60% of proteins above their LODs. Exceptions to this symbol color coding are the top five proteins identified in the three groups in **Fig 3B** for which the symbols have been individually enlarged, color-coded using the bar colors in **Fig 3B** (all were originally identified in the **GREEN** cluster), and labelled with their gene names. These highly correlating proteins are located at the extremes on the top, right and right upper sections of the plot (CD48 appears in both the HIV-1 and HIV-1 + NEFL groups and is colored according to the dual correlation). The dotted grid lines mark the 0 and 0.5 correlation levels for both the Y and X axes. Overall, the **GREEN** cluster proteins populate the highest NEFL and HIV-1 RNA correlation areas with a far larger number in the ≥0.5 of CSF NEFL correlation (Y axis) (375 proteins) than in the ≥0.5 area of CSF HIV-1 RNA correlation (X axis) (55 proteins). Twenty-two of these proteins are in the shared area in the upper right with correlations ≥0.5 for both CSF NEFL and CSF HIV-1 RNA. A minority of the GREEN proteins had <60% of their measurements above the LODs as indicated by the open green circles that exhibit NEFL correlations <-0.5 and CSF HIV-1 RNA correlations between 0.25 and 0.5. None of the 294 **BLUE**-clustered proteins are in the ≥0.5 HIV-1 RNA correlation area, and indeed most exhibit negative correlations with HIV-1 RNA; 52 of the **BLUE**-clustered proteins are in the ≥0.5 NEFL correlation area. Only a small number of the **BLUE**-clustered proteins show <60% of measurements above the LODs. Of the 566 **RED**-clustered proteins 10 are in the ≥0.5 correlations with NEFL and none correlate with HIV-1 RNA ≥0.05. These **RED**-clustered proteins thus are concentrated along the vertical line correlation designating a 0.0 correlation with CSF HIV-1 RNA. Additionally, many the **RED**-clustering proteins have <60% of proteins above the LOD (open symbols), likely contributing to their lack of correlations with either CSF NEFL or CSF HIV-1 RNA (with most grouped along the X axis at 0 correlation with CSF HIV-1 RNA). This figure with the 'cloud' of protein placement in relation to the two pathogenetic vectors emphasizes that the correlations of most of the Olink-measured proteins with CSF NEFL and HIV-1 RNA are largely 'dissociated'. Only a limited number lay along the diagonal dotted line delineating equal correlations with the two major disease variables. Additionally, the much larger number of proteins correlating with NEFL at levels above 0.5 than with HIV-1 RNA, emphasizes the major influence of CNS injury on the CSF proteome. This is consistent with the strong effect of **HAD** and **NSE** on CSF proteins as can be seen by the examples below and with many of the proteins identified in other sections of the study. By contrast, high HIV-1 RNA levels in some of the CD4-defined groups with normal CSF NEFL values also contributes to this dissociation. **D**. *Patterns of high-correlating proteins across subject groups*. This Panel shows the patterns of CSF protein concentrations of the top five proteins in the three categories identified in **Fig 3B**. Again, the dashed horizontal lines are at the mean + 2SD of the HIV-1- group as a visual reference for comparison of group value distributions. The *top row* shows the five proteins with the highest correlations with HIV-1 RNA. Here the highest protein concentrations vary between the *HAD* and *NSE* groups while the *CD4-defined groups* exhibit the *lymphoid* pattern with a reduction in the protein levels in the *CD4 <50* group compared to the *CD4 50–199* and *200–350* groups. This is not surprising because this is the pattern of the CSF HIV-1 RNA concentrations and CSF WBC counts in these groups (see **Fig 1I and 1K**). The patterns in the middle row of proteins with highest correlations with NEFL exhibit two major features: highest concentrations in the *HAD* group and a gradual increase across the *CD4-defined* groups with highest values within these in the *CD4 <50 group*, i.e., with the *myeloid* pattern. This circumstantially supports the association of CNS injury with myeloid cell inflammation. The final row with highest combined HIV-1 and NEFL correlations shows a mixture of the two patterns just outlined: variability in whether *HAD* or *NSE* had highest concentrations and in the presence of either the *lymphoid* or *myeloid* patterns in the *CD4-defined* groups. The identified proteins in **Fig 3D** are listed in **S1 Appendix** with brief functional descriptions extracted from UniProt database (https://www.uniprot.org/uniprotkb?query=*)[140].

In **Fig 3C** the five highest correlating CSF proteins identified in the three bar graphs in **Fig 3B** are highlighted with color-coding of their locations in the plot. All of these are inflammatory proteins, and, in a broad sense, their locations emphasize the major effect of CNS injury on CSF proteins, including particularly inflammatory proteins. However, the plot also shows that HIV-1 RNA measured in CSF is associated with protein changes that had a variable, often dissociated, relation to injury. Indeed, the highest correlations with CSF HIV-1 RNA are infrequently highly correlated with the extent of CNS injury as measured by CSF NEFL and, vice-versa, proteins correlating strongly with neuronal injury are, in general, relatively weakly correlated with HIV-1 RNA. These dissociations also provide exploratory maps for examining pathogenic relationships, including defining inflammatory increases with weak and strong relationships with CNS injury and CSF HIV-1 RNA.

**Fig 3D** shows the group CSF profiles for the five CSF proteins with highest correlations with HIV-1 RNA, NEFL and HIV-1 + NEFL, respectively, that were identified in **Fig 3C**. The patterns of change in the proteins in the first two groups show consistent differences. Those with high HIV-1 RNA correlations (top row) exhibit the *lymphoid* pattern across the *CD4+ T-cell-defined groups* while the NEFL-correlating proteins (second row) more closely followed the *myeloid* or perhaps *neuronal* pattern of change across the *CD4-defined* groups with a gradual or more abrupt increase to highest levels in the **CD4 <50** group. There are also evident variations in these patterns among these proteins; for example, PLAUR on the far right more closely resembles the NEFL's *neuronal* pattern. In the two upper rows of graphs in **Fig 3D** the protein levels are more often higher in the *HAD* group than in the *NSE* group, but not in all. The combined HIV-1 and NEFL examples (third row) exhibit a mixture of the patterns seen in the other two groups (e.g., CD48 with the *lymphoid* and LILRA5 and LGALS9 with the *myeloid*

patterns). Of note, the **RxNFL+** group does not show elevation in the inflammatory proteins that associate with NEFL, suggesting that NfL elevations in this subgroup are likely due to different processes than those involved in the two forms of **HIVE**. This is also in agreement with the result of the other neural injury marker measurements discussed earlier and shown in S2 *Fig*.

Overall, these results emphasize that the correlations of CSF proteins with CSF NEFL and HIV-1 RNA are frequently *dissociated*, and that, indeed, the proteins with the strongest correlations with CSF NEFL are more commonly associated with relatively weak CSF HIV-1 RNA correlations and, conversely, strong correlations with CSF HIV-1 RNA are frequently accompanied by relatively weaker correlation with CSF NEFL. Additionally, high correlations with NEFL associate with *myeloid* (and perhaps *neuronal*) patterns of change across the **CD4-defined** groups, while high correlations with HIV-1 RNA associate with the *lymphoid* pattern of change across these groups. From this, we can speculate that the proteins correlating strongly with CSF HIV-1 RNA are mainly from T-cells reacting to or even enhancing local HIV-1 production or replication, while the proteins correlating strongly with NEFL are either effecting or reacting to CNS injury and involve virus populations that evolved to grow in CNS myeloid cells, though we have not directly established these immune response-virological linkages and this first analysis does not distinguish between inflammatory biomarkers that may have different levels of importance in **HAD** and **NSE**.

## CSF proteins differences between *HAD* and *NSE*

In addition to characterizing the broad changes in the CSF proteome across the defined clinical groups, this dataset can be used to explore features that distinguish individual clinical groups or sets of groups from each other. As an initial example of this application, we next compared the CSF proteins of the two groups with major HIV-1-related CNS injury: the **HAD** and **NSE** groups. Among the distinguishing background features of these two groups with **HIVE** are: i) the absence and presence of treatment (with full or partial plasma viral suppression); ii) consequent difference in blood CD4+ T-cell counts (median of 108 cells per μL in **HAD**, and 425 cells per μL in **NSE**); iii) difference in CSF HIV-1 RNA levels (median of 5.27 $\log_{10}$ copies per mL in **HAD** and 3.25 $\log_{10}$ copies per mL in **NSE**); iv) CSF WBC counts that were elevated in both but twice as high in **NSE**; and v) different CSF NEFL concentrations (higher in **HAD**). Previous studies also have shown that the pathology and HIV-1 genotypes/phenotypes of CSF isolates also usually differ. **HAD** is typically a multinucleated-cell encephalitis [129,150,151] involving myeloid cell infection with M-tropic viruses [152–154] developing in untreated individuals with advanced systemic infection, while **NSE** is generally a T-cell mediated disease with notable CD8+ T-cell infiltration pathologically [102] and T-tropic HIV-1 infection [155]. NSE usually occurs in the setting of limited CNS penetration of one or more antiretroviral drugs along with resistance to other drugs [94–96,101,103,104,131]. **Fig 4** compares the CSF proteins of these two types of **HIVE**.

This figure compares the CSF proteins in **NSE** and **HAD**. While the impetus for this comparison was largely to identify protein differences, in fact, we found that the great majority of the CSF proteins were not significantly different in this analysis and that they clustered at the base of the Volcano Plot in **Fig 4A**. On reflection, the lack of major differences may not be surprising because both conditions are highly inflammatory types of **HIVE**, and whatever the root cause of virus- or inflammatory-related neuronal injury, the accompanying responses may be similar However, there are some distinctions as shown by the labelled proteins in **Fig 4A** and the bars in **Fig 4B** that identify several proteins that are more prominent in one or the other condition. **Fig 4C**, examining the correlations of these proteins with the two 'pathogenic

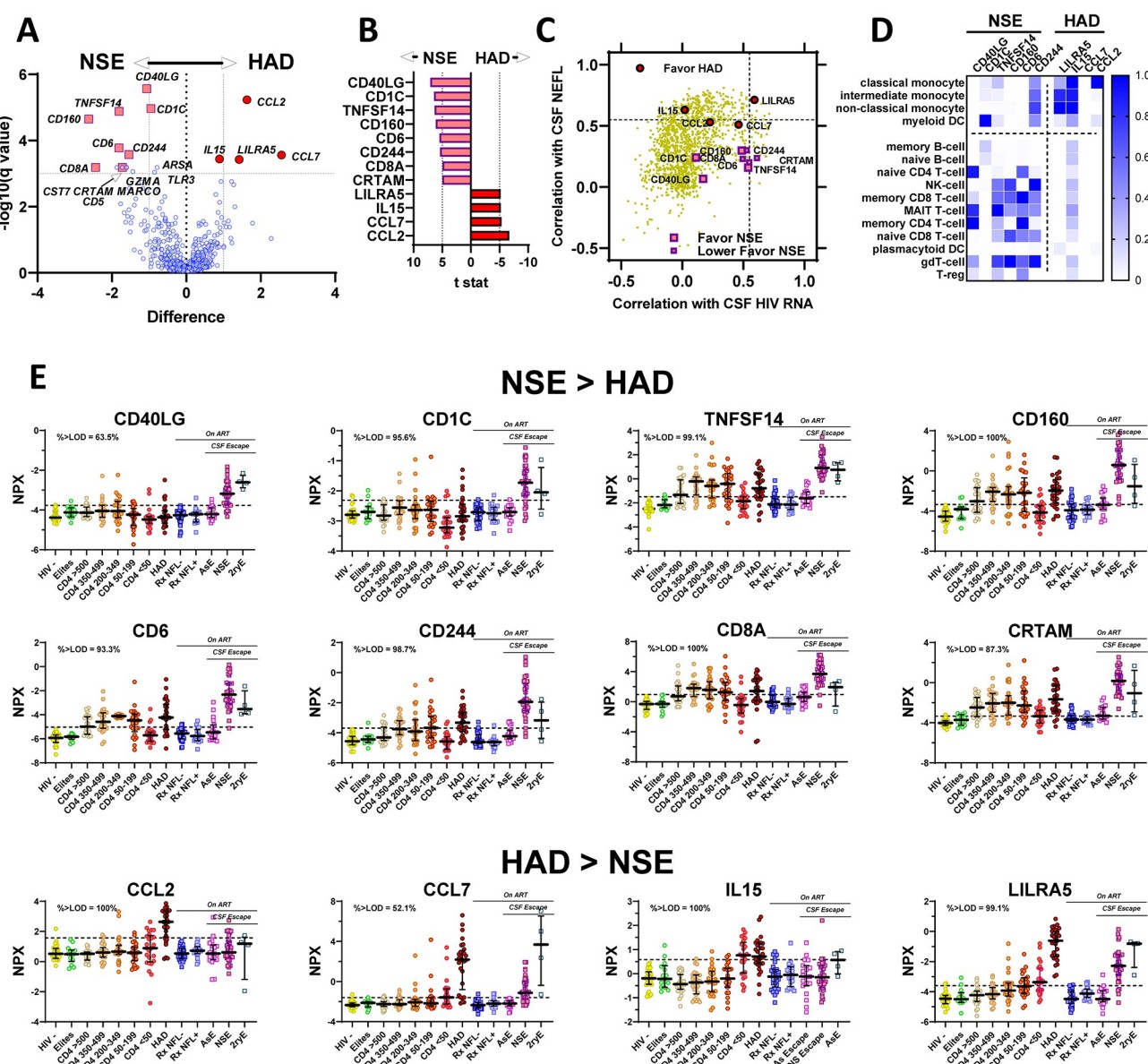

**Fig 4. CSF proteins favoring *NSE* vs *HAD*.** This figure examines CSF protein differences between *NSE* and *HAD*. **A. Comparison of CSF proteins in *NSE* and *HAD* groups.** Both of these conditions are forms of *HIVE* with elevations of multiple CSF proteins. For the most part, the protein elevations do not differ significantly, and this volcano plot shows a high degree of protein similarity (or lack of significant differences) within the context of the full complement of measured proteins (identified by simple blue symbols) on each side of the central vertical axis. However, there are a few identified protein differences, indicated by the enlarged labelled symbols formatted using the *NSE* and *HAD* symbols. The remaining panels of the figure focus on these distinguishing proteins. **B. CSF proteins distinguishing *NSE* and *HAD* subject groups.** Proteins identified by **t stat** of ≥ +/- 5 are the same those singled out in **Fig 4A** volcano plot, with eight proteins favoring *NSE* and four favoring *HAD*. **C. NSE and HAD distinguishing proteins in the context of their correlations with CSF NEFL and CSF HIV.** This panel shows all the measured Olink proteins (except NEFL) in the context of their correlations with CSF NEFL and CSF HIV RNA using the same format as *Fig 3C*, but, for clarity, with a neutral yellow color for the full array of proteins rather than the **GREEN**, **BLUE** and **RED** designations from the hierarchical cluster analysis. The significant proteins identified in the volcano plot and t stat above (**Fig 4A** and **4B**) are indicated by enlarged symbols. All are from the **GREEN** group of the cluster analysis. The proteins favoring both neurological conditions are similarly variable across a range of correlations with CSF HIV-1 RNA but located in different vertical strata with respect to NEFL correlation. Those favoring *HAD* are in a higher range of correlation with NEFL than those favoring *NSE*. This may relate, at least in part, to the generally lower levels of NEFL in *NSE* than *HAD* (**Fig 2A**). **D. Cellular associations of proteins favoring NSE and HAD.** This simple presentation draws on literature sources describing the cell associations of 10 of the proteins identified in *Fig 4A* and *4B*, https://www.proteinatlas.org/ and [156] with the darkness of the square indicating strength of the cell relations. The proteins listed as favoring *NSE* are largely associated with lymphocytes (B, T and NK lineages), while those favoring *HAD* associate with myeloid cells, consistent with a different balance of inflammatory cells in these two forms of *HIVE*. **E. Different patterns of high-correlating proteins in NSE and HAD across subject groups.** The format of the graphs in this panel is the same as in previous figures showing Olink protein concentrations across the subject groups. The upper two rows plot the protein concentrations of the 8 proteins

favoring *NSE* while the bottom row plots the proteins favoring *HAD* as identified in *Fig 4A* and *4B* above. In the upper two rows, the median concentrations in the *NSE* group are greater than those of the comparable *HAD* group which are either normal or near the highest values among the *CD4-defined* groups. The *CD4-defined* groups exhibit the *lymphoid* pattern with varying levels of increase in the middle brackets. The bottom row shows the patterns of the four proteins that favor *HAD*. Most notably, the values of *HAD* group are higher than those of *NSE* group, and the *CD4-defined* group patterns are either flat or show an increase in the *CD4<50* group above the groups with higher CD4+ T-cell counts, closest to the *myeloid* or perhaps *neuronal* patterns, but with clear variation in the magnitude of differences of the *CD4<50* from the other *CD4-defined* groups. The dashed horizontal line in each panel again designates the mean +/- 2 SD of the *HIV-* control group for visual reference. The identified proteins in **Fig 4E** are listed in **S1 Appendix** with brief functional descriptions extracted from UniProt database (https://www.uniprot.org/uniprotkb?query=*) [140].

vectors' introduced earlier shows that these proteins both stretch across the spectrum of correlations with CSF HIV RNA but appear in different strata with respect to correlation with CSF NEFL so that those favoring *NSE* show lower and those favoring *HAD* have higher correlations with this axonal injury protein. In part, at least, this likely relates in some way to the lower concentrations of CSF NEFL in the treated *NSE* patients, but **Fig 4D** using literature data suggests that these proteins relate predominantly to different cells, *NSE* to *lymphoid* and *HAD* to *myeloid* inflammatory processes. This is also consistent with the profiles of these proteins across the subject groups shown in **Fig 4E** in which those favoring *NSE* not only show higher concentrations in the *NSE* group than that of *HAD*, but also show the *lymphoid* pattern while those favoring *HAD* exhibit a *myeloid* or *neuronal* pattern across the *CD4-defined* groups. Thus, overall, the inflammatory components of these two types of *HIVE* exhibit considerable similarity, but also some discernable differences.

These initial comparisons are all consistent with the neuropathological differences between the two forms of *HIVE*, with *HAD* typically a myeloid-cell inflammatory and M-tropic virus-dominated process, and *NSE* mainly featuring lymphocytic infiltration and infection [102,157] in association with infection by presumed T-tropic viruses. To what extent these CSF proteins simply reflect reactions to the CNS injury or, conversely, importantly participate in driving it, remains uncertain. Are the main drivers of CNS injury in these two conditions similar but the inflammatory reactions vary because of differences in the background immune system reactive capabilities, or are the mechanisms of injury more fundamentally different?

## Conclusions and future directions

This study generated an abundance of novel data related to the robust and complex changes in CSF proteins over the course of HIV-1 infection, including after treatment. The full dataset is posted so that other investigators can now join in extending the analysis of these data. This initial report is largely aimed at introducing the study and its rich dataset. It pursues an agnostic, largely empirical approach to discovering and outlining the contours of CSF protein changes over the full course of chronic HIV-1 infection. Future studies now need to use these data to examine mechanisms and unravel underlying pathogenetic issues. These might be approached from at least two directions: **1**. Continued comparisons of subject groups or subgroups to characterize and understand the distinct features of each and how they differ from other groups in order to map transitions and salient differentiating features of the various stages of infection and injury; and **2**. Informed analysis of the changes in particular proteins, in related proteins and in established protein networks and pathways to examine the mechanisms underlying the changing CSF proteome features at these different phases of systemic and CNS HIV-1 infection. CSF inflammation is a nearly ubiquitous feature through the course of infection, and likely a central pathogenetically facet of CNS HIV-1 infection. Its relation to both systemic infection and CNS injury are clearly key aspects of CNS infection and its consequences. Further definition and understanding of these CSF protein changes promises to provide enhanced understanding of the CNS pathobiology of HIV-1 infection and perhaps additional approaches

to more precise diagnosis and to neuroprotective and neuro-mitigating therapies. This study also demonstrates the potential value of applying a similar approach, including extension to measurement of non-protein molecules, to discovery aimed at pathogenetic and diagnostic exploration of other infectious, inflammatory and degenerative neurological diseases.

## Supporting information

**S1 Table. Study group background characteristics.** This file contains a table that summarizes the salient background characteristics of the study groups.
(XLSX)

**S1 Appendix. CSF in HIV (one file uploaded separately).** This appendix contains brief glossaries relating to the proteins identified in the figures.
(DOCX)

**S1 Fig. Quadruplicate measurements of CXCL8, IL6 and TNF.** This figure examines the correlations among quadruplicate assays of three proteins and other selected features including QC warnings and LOD measurements of inflammatory molecules. The Olink Explore 1536 platform uses four individual assay plates (*Cardiometabolic*, *Inflammation*, *Neurology* and *Oncology*), each with sets of protein detection antibodies (two antibodies for each protein measured). As part of the quality control strategy, three of the proteins (CXCL8, IL6 and TNF) are measured on all four plates, i.e., in quadruplicate, using the same antibody pairs but in the context of different sets of background antibody pairs on each of these plates. **S1 Fig** examines the intercorrelations of the assay results and compares the patterns of study specimen profiles generated by each of the quadruplicate assays across the spectrum of the 307 study specimens (this preliminary exploration included all of the specimens, including the four miscellaneous samples omitted from the main analysis). Additionally, the plates are structured so that assay of each protein is divided into four component sets of measurements, each with a separately calculated level of detection (LOD). Examination of the three sets of quadruplicate measurements in **S1 Fig** provides an opportunity to concretely illustrate these methodological features which are annotated for every protein in the full data table (**File 1 Dataset of CSF proteins in chronic HIV _infection_rev2024-09-10.xlsx DOI: 10.5061/dryad.x3ffbg7tv**). **A-C. Correlations among quadruplicate plate assay results**. These panels include three montages, each with 12 graphs showing assay correlations among the quadruplicate assays with identification of the QC Warnings related to individual measurements and notations of assay LOCs. The format for the first set (CXCL8) is outlined in more detail that can then be applied to the other two sets of quadruplicates (IL6 and TNF), while the LOD notations are more prominent in the second two sets, and particularly in the third (TNF). *A. Correlation of CSF CXCL8 measurements in quadruplicate assays*. This panel shows the correlations of the results of four assays (one from each plate) of the CSF CXCL8 concentrations in NPX units. The assay result numbers and plate names in the panels without graphs refer to the protein results graphed on the X axis of the panels in that row, and also to the Y axis of the panels in the column, e.g., measurement 20153 was obtained on the Cardiometabolic plate and these results were used in the X axis of the other graphs in the (top) row and the Y axis of the graphs in the (left) column. The panels show both the close correlations of the results across the quadruplicate measurement and the variability of absolute results among the plates and indeed of the scales of results among plates. The graphing of the points with QC warnings shown in orange relate to the measurement shown in the column of the panel and to the protein on the Y axis (for example, the QC Warnings for the Cardiometabolic plate warnings for its CXCL8 assay are included in the three graphs below this label). With respect to the correlations, there are actually only 6

sets of correlations, but the other 6 panels show graphs of the same variable pairs mirrored across the diagonal of the panel names. These graphs show the QC Warnings of different assays (those of the columns). The six linear correlations performed across these results are all very tight with an overall mean R2 = 0.9821 +/- SD 0.0059. The "*" on the upper left box (20153 assay of CXCL8 on the Cardiometabolic plate) indicates that the results for the CXCL8 measurements from this plate were selected at random prior to analysis and used in the analyses presented in this report (and highlighted by gold-yellow background in **File 1 Dataset of CSF proteins in chronic HIV _infection_rev2024-09-10.xlsx** available on-line Dryad (DOI: 10.5061/dryad.x3ffbg7t). All of the QC Warning samples are within the body of other assay results. The panels do not show the LOD levels for these assays since all are well below the range of all the measurements. ***B. Correlation of CSF IL6 measurements in quadruplicate assays***. This panel, showing the correlations among the four CSF IL6 assays (**S1B Fig**) is arranged in the same format as the CXCL8 panel discussed above and uses the same labelling rules. Again, the correlations among the 6 assays are high with a mean $R^2$ mean of 0.9508 +/- SD 0.0142. The measurements with QC warnings are also again all well within the body of the overall data. The graphs also include the LODs for each of the four assays plotted on both the X and Y axes. These intrude on some of the lowest IL6 values but likely have little impact on the overall interpretation of the correlations or differences among the groups (see **S1D Fig** below). ***C. Correlation of CSF TNF measurements in quadruplicate assays***. This panel is also arranged in the same format as the CXCL8 and IL6 results discussed above. However, the TNF measurement correlations among the quadruplicates are distinctly lower: the mean $R^2$ was 0.7055. Additionally, two of the measurements with QC Warnings are outliers; this is most easily visualized in the two graphs in the Oncology plate results column (middle two subpanels on the right). Finally, the LOD plots show that a substantial number of the values are below the four LODs. The impact of this is also seen more clearly in **S1D** below. ***D. Subject group profiles of the three sets of quadruplicate CSF protein measurements across the study subject groups***. This group of graphs displays the results of each of the 3 quadruplicate assays (CXCL8, IL6 and TNF in the top, middle and bottom rows of graphs, respectively) for all the subjects in this analysis. They are derived from the same measurements as the above panels but displayed according to the subject groups using the same format (including color coding of symbols and median +/- SEM error bars) for other group profiles in this report, as outlined in **Fig 1**. In each of these panels the horizontal dashed line is set at the mean + 2SD of the **HIV negative** controls for visual reference across the 13 groups while the four dotted lines indicate the four LOD values for the individual assay. The patterns of CSF protein changes across the groups are very similar for each of the quadruplicate measurements of the individual proteins (CXCL8, IL6 and TNF), including in relation to the **HIV-** control values, changes in medians across the **CD4-defined** groups as blood CD4 cells fell, elevations of **HAD** and **NSE** relative to each other (for example: modest elevations of CXCL8 with median values of **NSE > HAD**; elevations of IL6 in **HAD** with relatively low values for **NSE**; and nearly equal elevations of both **HAD** and **NSE** for TNF). While for each of the four replicates the relative changes in the subject groups are largely the same, there are also notable differences across some of the assays in the absolute NPX values and ranges. For example, the NPX ranges and values in the four panels of CXCL8, particularly between the right and left panels (Cardiology and Oncology results), differed. This is less marked for the IL6 repeats, although there are notable the differences in the **HIV-** medians in the left two panels of TNF results (Cardiology and Inflammation). For TNF, there also are differences in the mean values among the four measurements. These comparisons thus clearly illustrate that the NPX measurements report relative concentrations within each assay that can differ among the quadruplicates, yet show similar relative changes with respect to group comparisons. The figure also shows the four LOD values for each of the assays with

dotted horizontal lines. In the case of CXCL8 (top row of four graphs), these are all well below all the sample values; thus, results of this assay are not impacted by low values below the LODs. For IL6 (middle row), a few of the lowest values are near or below the LODs, but this likely had little influence on the patterns of change or correlations with other biomarkers. By contrast for TNF (lowest row) the LODs are at or above the medians of the *HIV-* controls and several of the other groups with 25.7 to 69.2 percent of measurement results above the LODs for this protein. While the median *HAD*, *NSE* and *2ryE* groups clearly show elevated TNF values in each assay, the lower values for most of the other groups that are near or below the LODs clearly interferes with interpretation and comparison of the overall pattern of change for TNF. (PDF)

**S2 Fig. CSF protein biomarkers of CNS injury measured in Olink Explore platform.** The figure shows the results of seven **Olink Explore 1536** protein measurements that have been applied to assess CNS injury in other studies. In **S2A** and **S2B-H Fig** the format is the same as that for other CSF biomarker changes across the groups, including the horizontal dashed line that indicates the mean + 2 SD of the *HIV-* controls as a visual reference for comparison across group (or– 2 SD in the case of APP in **S2D Fig** in which the concentration changes in the PWH were negative). The percentage of measurement above the LODs for each of these is also listed, with all in a range that allows interpretation of the values across the subject groups. Following the nomenclature used in the **Olink Explore 1536** platform, the proteins are designated by their gene names. This includes the Olink measurement of NfL. In this paper we use the abbreviation, *NEFL*, for the Olink results while using *NfL* as the abbreviation for general references to this protein and to the results of other assays, Including the UMAN ELISA used for the earlier measurements performed in the University of Gothenburg Laboratory of Neurochemistry that are presented in [Fig 1O]. The following sections briefly discuss each panel in the figure. **A. The NEFL measurements generated in the Olink Explore 1536** exhibit a pattern of concentration changes very similar to that seen with the UMAN ELISA results presented earlier in [Fig 1O]. Greatest elevations are present in the *HAD* group, minor increases in the *CD4 <50* group, and substantial elevations in the *NSE* group (though lower than in *HAD*). The *Rx NFL+* group shows clear elevation with a median near the level of the HIV- mean + 2 SD dashed line. This pattern might be considered as the prototypic *neuronal* pattern of change across the sample group. It is similar to the *myeloid* pattern, perhaps distinguished by the flat concentration levels in the *CD4-defined groups* with blood CD4+ T cells above 50 per μL, along with elevation in the *NSE*. Whether the general elevation in the *Rx NfL+* group should be considered a part of this pattern or is peculiar to NfL alone requires further understanding of this group. Clearly minor distinctions from the *myeloid* pattern require further analysis and validation. **B. Correlation of CSF NEFL (Olink) with prior NfL measurements without age adjustment.** The regression shows the high correlation ($R^2$ = 0.8790) between the Olink NEFL and the prior Uman ELISA assay results in the subset of the subjects presented in [Fig 1O] but without age adjustment. In this comparison the high correlation is noted despite the earlier measurements having been performed in multiple separate laboratory runs over several years. As in **S1 Fig** comparing analysis of the quadruplicate repeated assays, the NEFL measurements with QC Warnings, shown by orange symbols, fall within the central body of the results along the regression line, further supporting the inclusion of measurements with QC Warnings in our analysis. **C. MAPT** (P10636). In this figure, only the *HAD* group exhibits a clear median increase with about half of the results above the 2 SD level of the *HIV-* control mean. The median of the *NSE* group is similar to that of the control group with a tail of about 25% of the samples falling above the 95% confidence limit. Hence, this protein appears to be a less sensitive marker of neural injury in HIV-1 infection. **D. APP** (P05067). Measurement of this

protein in the panel shows only a modest decrease in median level in the **HAD** group and a seeming lower median in the **2ryE** group. Thus, both MAPT and APP CSF measurements, which serve as indicators of Alzheimer's disease pathology, are not sensitive to HIV-1-related CNS injury. **E. GFAP** (P14136). CSF GFAP is normal in most of the groups except for **HAD** and to a lesser extent in **NSE** with about 50% and <50% concentrations above the controls +2 SD line, respectively. Either GFAP is a relatively insensitive biomarker of astrocytic reactions in this setting, or astrocytic changes are a less prominent aspect of HIV-induced neuropathology than neuronal injury (as documented by CSF NEFL changes). **F. CHI3L1/YKL-40** (P36222). CHI3L1/YKL-40 concentrations in CSF are distinctly elevated in HAD and NSE with an overall pattern similar to that of NEFL, but with an earlier median increase in the *CD4-defined* groups. Incidentally, the value in the **Rx NFL+** group is similar to that of the **Rx NFL-** group, which might imply that if injury is present in the **Rx NFL+** group, it is circumscribed, affecting axons but sparing other cellular elements. The small **2ryE** group values are also elevated, presumably related to strong glial reaction in these individuals with a *myeloid* pattern in the **CD4-defined groups**. Overall, CHI3L1/YKL-4 appears to be a sensitive biomarker of HIV-induced pathology. **G. TREM2** (Q9NZC2). In this study, there was an increase in TREM2 mean values in the lower **CD4-define**d groups and highest levels in the **HAD** group, but largely normal levels in **NSE** except for some elevated outliers possibly indicating a difference in their pathological substrates. This is consistent with its putative association with microglial activation and a myeloid predominance in **HAD** but not in **NSE**. The gradual increase in its concentration with falling CD4+ T cells, including the **CD4 <50** group, also shows the *myeloid* pattern. This differences from CHI3L1/YKL-4, perhaps most notably the absence of elevation in the NSE group suggests that these two markers detect different cell changes, with YKL-40 importantly encompassing astroglial changes present in both **HAD** and **NSE**, while TREM2 largely indicates myeloid activation that is more prominent in **HAD** than in **NSE**. **H. KYNU** (Q16719). (*Kynureninase*). There was a prominent increase in **HAD** and lesser increase in **NSE** with little change from control in the other groups. The pattern in **CD4-defined** groups was perhaps ambiguous, but the overall median values of these groups were close to that of the HIV- controls. KYNY might be a useful marker of severe CNS injury. The highest correlations (P <0.0001) of these markers with NEFL included: CHI3L1 ($R^2 =$ 0.6005), TREM2 ($R^2 = 0.5622$), GFAP ($R^2 = 0.4545$) and MAPT ($R^2 = 0.3471$). (PDF)

## Author Contributions

**Conceptualization:** Richard W. Price, Magnus Gisslén.

**Data curation:** Zicheng Hu, Shuntai Zhou, Richard W. Price, Magnus Gisslén.

**Formal analysis:** Zicheng Hu, Shuntai Zhou, Richard W. Price, Magnus Gisslén.

**Funding acquisition:** Richard W. Price, Magnus Gisslén.

**Investigation:** Paola Cinque, Ameet Dravid, Lars Hagberg, Aylin Yilmaz, Henrik Zetterberg, Dietmar Fuchs, Johanna Gostner, Serena S. Spudich, Richard W. Price, Magnus Gisslén.

**Methodology:** Richard W. Price, Magnus Gisslén.

**Resources:** Paola Cinque, Ameet Dravid, Lars Hagberg, Aylin Yilmaz, Richard W. Price, Magnus Gisslén.

**Supervision:** Ronald Swanstrom, Richard W. Price, Magnus Gisslén.

**Visualization:** Zicheng Hu, Richard W. Price, Magnus Gisslén.

**Writing – original draft:** Zicheng Hu, Richard W. Price, Magnus Gisslén.

**Writing – review & editing:** Zicheng Hu, Paola Cinque, Ameet Dravid, Lars Hagberg, Aylin Yilmaz, Henrik Zetterberg, Dietmar Fuchs, Johanna Gostner, Kaj Blennow, Serena S. Spudich, Laura Kincer, Sarah Beth Joseph, Ronald Swanstrom, Richard W. Price, Magnus Gisslén.

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
