## [Decision Letter · Decision Letter 0]

24 May 2024

Dear Dr. Price,

Thank you very much for submitting your manuscript "Changes in Cerebrospinal Fluid Proteins 1 across the Spectrum of 2 Untreated and Treated Chronic HIV-1 Infection" for consideration at PLOS Pathogens. As with all papers reviewed by the journal, your manuscript was reviewed by members of the editorial board and by several independent reviewers. In light of the reviews (below this email), we would like to invite the resubmission of a significantly-revised version that takes into account the reviewers' comments.

We cannot make any decision about publication until we have seen the revised manuscript and your response to the reviewers' comments. Your revised manuscript is also likely to be sent to reviewers for further evaluation.

Sincerely,

Guido Silvestri

Academic Editor

PLOS Pathogens

Richard Koup

Section Editor

PLOS Pathogens

Michael Malim

Editor-in-Chief

PLOS Pathogens

orcid.org/0000-0002-7699-2064

Reviewer's Responses to Questions

**Part I - Summary**

Reviewer #1: The authors used proteomic profiling to characterize 303 CSF specimens from 12 groups of participants from 4 clinical sites. The signatures identified confirmed two previous patterns observed in previous studies using only a few markers including the lymphoid (inflammatory changes that increase with decreasing CD4 count until decreasing when below 50 cells/uL) and myeloid (increasing levels with deceasing CD4 count). Antiretroviral treatment mostly mitigated the changes but not necessarily to control levels. The authors have generously made the data fully available for others to collaborate. It would be helpful to also have some of the key neurocognitive scales and reservoir measures to further contextualize the findings. Additional considerations are included here.

Reviewer #2: This is a comprehensive, thoughtful study of cerebrospinal fluid (archived specimens) protein expression in 303 patients with HIV (PWH) that include ART-treated and untreated PWH through mild and severe neurocognitive impairment (HIV-assosciated dementia). Uninfected control patient samples were also included, as were samples from patients with neurosymptomatic and asymptomatic CSF escape (NSE, ASE) . This is a non-biased sampling of more than 1400 proteins, and this study is unique in its relatively large size and its detailed information on the clinical and functional status of the PWH. Additionally, the examination of NSE is a unique feature. It is rigorously performed. The response to ART adds yet another important piece of information to this comprehensive study.

The authors acknowledge controversies in the field, and they thoughtfully frame points worthy of more investigation and discussion, some of which may be ignored by researchers and clinicians. The background information is clearly presented, up-to-date and comprehensive. They acknowledge the limitation of previous studies (including their own), and they appropriately emphasize the limitations of associations of markers with clinical status and pathways as not distinguishing between driving or contributing to pathogenesis or rather responding to pathogenic processes. The conclusions are strongly supported, and they are not over-reaching.

The findings of responses associated with lymphoid and/or myeloid lineages is novel, and strongly consistent with the entry and replication of lymphocyte-tropic, non-M-tropic strains into the CNS with subsequent evolution of M-tropism, establishment of a myeloid reservoir and possibly a lymphoid reservoir (?meninges). The association of CSF Nfl levels with a myeloid pattern of marker expression and not with CSF vRNA levels (which associate with lymphoid marker pattern) is an enlightening observation. Also HAD indeed associates with a myeloid pattern while NSE associates with a lymphoid pattern. These observations and interpretations are consistent with our knowledge of the entry into and evolution within the CNS of HIV strains. Overall, this study greatly enhances our knowledge and potential understanding of the CNS infection process and consequences thereof.

Minor points:

1) It would be helpful to speculate more about the role of CSF or plasma Nfl quantification in judging the degree of neuronal injury in less severely impaired patients. It appears that it may not really be useful. How helpful is it in cases other than HAD? As it is now clinically available, should plasm Nfl be routinely monitored in PWH? In selected patients?

2) The study involves analyses of CSF proteins, as have many smaller studies previously. Some comment about other pathogenic moieties (lipid peroxidation products, ROS, mtDNA, metabolites, others) that might also be relevant to pathogenesis can further help the reader to appreciate the complexity of neuropathogenesis.

3) Figure 2 includes data from HIV- and elite controllers----this is defined appropriately, but it should probably be introduced in the background section. Are there enough data to speculate about distinctions between elite controllers and virally-suppressed PWH?

**Part II – Major Issues: Key Experiments Required for Acceptance**

Reviewer #1: 1. Line 125 – the authors should explain if the patterns of “CSF inflammatory reactions”, which are compartmentalized and evolve independently, appear different from those in the blood.

2. Line 128 – the authors should clarify if the phrase “augmented by additional inflammatory biomarkers” meant that the additional markers had the same patterns but were higher or just had the same patterns.

3. Line 165 – I would clarify that this relationship is potentially bidirectional.

4. Line 183 – since these are cross-sectional assessments, the authors should comment as to whether the AsE phenotype could potentially be an earlier version of the NSE phenotype

5. Line 195 – similar to the point above the authors should state if there are instances where any participants sampled as an AsE or higher CD4 group belonged to the NSE or lower CD4 group at a later time, respectively.

6. Methods – it is not clear where individuals with asymptomatic neurocognitive impairment and mild neurocognitive disorder would appear with or without escape or whether there would be individuals without any impairment but had escape.

7. Line 248 – it is not clear if any of the CD4 category participants could have phenotypic overlap with the NSE or HAD participants.

8. Line 263 – the authors should explain why sCD14, as a monocyte marker, follows the pattern associated with lymphocyte reduction.

9. Line 291 – the authors should explain why the CSF:blood ratio is different between NSE/2ryE vs HAD if neopterin is only produced in the CNS.

10. Line 345 – the authors should describe how the LOD levels were imputed.

11. Line 413 – the separation was less than I would have imagined and would have thought that HAD would be the lowest; the authors should address this.

12. Line 431 – the authors should address if any of these patterns suggest a mechanistic difference with these two phenotypes (HAD and NSE).

13. Line 486 – it is not clear why log of the HIV VL or some kind of weighted sum was used versus a straight sum of these two measures.

14. Line 637 – the CD4 threshold criteria should be described.

15. Line 642 – the HIV plasma viral load and ART use criteria should be described.

Reviewer #2: No additional experiments needed

**Part III – Minor Issues: Editorial and Data Presentation Modifications**

Reviewer #1: 1. Line 105 – I would consider changing to the more common “R5 viruses” instead of “M-tropic viruses”.

2. Line 110 – I would consider changing “morbidities” to “co-morbidities”.

3. Line 114 – would change “T cell” to “T-cells”.

4. Line 118 – “…this ‘quadratic’ pattern of CSF viral load change…”, please clarify what this change is as it is not described or mentioned earlier.

5. Line 242 – the authors state that the CD4 groups were introduced earlier but a definition was not previously given (appears later on line 248).

6. Line 273 – the authors should mention which group was greater between NSE and 2ryE.

7. Line 276 – RxNFL+ and RxNFL- should be defined.

8. Line 291 – the AsE group is not described.

9. Line 304 – the authors should speculate as to why the RxNFL+ group has an elevated NfL

10. Line 310 – should clarify if the relative concentrations are relative to HIV negative.

11. Line 314 – would explain why a random selection was done rather than average of the results.

12. Line 317 – explain whether the LOD is irrespective of HIV negative levels or after adjustment.

13. Line 367 – may need to add “of NfL” to “…whether the Olink measurement of NfL, which…”.

14. Line 371 – would also add “age” in front of second “adjusted” at the end of the sentence.

15. Line 472 – perhaps the “>” should be “<”.

16. Line 624 – would avoid “HIV infected” to refer to people or groups per the community request and in Line 680 change “subject” to participant.

17. Line 659 – it should be clarified if NSE impairment was mild, moderate or severe.

Reviewer #2: Minor points:

1) It would be helpful to speculate more about the role of CSF or plasma Nfl quantification in judging the degree of neuronal injury in less severely impaired patients. It appears that it may not really be useful. How helpful is it in cases other than HAD? As it is now clinically available, should plasm Nfl be routinely monitored in PWH? In selected patients?

2) The study involves analyses of CSF proteins, as have many smaller studies previously. Some comment about other pathogenic moieties (lipid peroxidation products, ROS, mtDNA, metabolites, others) that might also be relevant to pathogenesis can further help the reader to appreciate the complexity of neuropathogenesis.

3) Figure 2 includes data from HIV- and elite controllers----this is defined appropriately, but it should probably be introduced in the background section. Are there enough data to speculate about distinctions between elite controllers and virally-suppressed PWH?

PLOS authors have the option to publish the peer review history of their article (what does this mean?). If published, this will include your full peer review and any attached files.

Reviewer #1: No

Reviewer #2: No
---

## [Decision Letter · Decision Letter 1]

1 Aug 2024

Dear Dr. Price,

We are pleased to inform you that your manuscript 'Changes in Cerebrospinal Fluid Proteins across the Spectrum of Untreated and Treated Chronic HIV-1 Infection' has been provisionally accepted for publication in PLOS Pathogens.

Best regards,

Guido Silvestri

Academic Editor

PLOS Pathogens

Richard Koup

Section Editor

PLOS Pathogens

Michael Malim

Editor-in-Chief

PLOS Pathogens

orcid.org/0000-0002-7699-2064

Reviewer Comments (if any, and for reference):

Reviewer's Responses to Questions

**Part I - Summary**

Reviewer #1: (No Response)

Reviewer #2: This study is comprehensive and it rationally dissects the extensive data and links those data to the clinical features of HIV associated cognitive dysfunction. The authors are appropriately critical of their own data and studies and they present a balanced view of the same. Likewise for other published work in this area.

**Part II – Major Issues: Key Experiments Required for Acceptance**

Reviewer #1: (No Response)

Reviewer #2: No additional experiments necessary

**Part III – Minor Issues: Editorial and Data Presentation Modifications**

Reviewer #1: (No Response)

Reviewer #2: No additional issues---all addressed as requested

PLOS authors have the option to publish the peer review history of their article (what does this mean?). If published, this will include your full peer review and any attached files.

Reviewer #1: No

Reviewer #2: No

---

## [Editor Report · Acceptance letter]

13 Sep 2024

Dear Dr. Price,

We are delighted to inform you that your manuscript, "Changes in Cerebrospinal Fluid Proteins across the Spectrum of Untreated and Treated Chronic HIV-1 Infection," has been formally accepted for publication in PLOS Pathogens.

Best regards,

Michael Malim

Editor-in-Chief

PLOS Pathogens

orcid.org/0000-0002-7699-2064